# SOX2 confers tumour permissiveness in a specific skin progenitor population

Patricia P. Centeno [1,2] ✉, Christopher Chester[2,4], Georgios Kanellos [1], Catriona A. Ford[1], Patrizia Cammareri[1,5], Gareth J. Inman [1,3], Thomas Jamieson[1], Rachel A. Ridgway[1], Richard Marais [2,4], Andrew D. Campbell [1] & Owen J. Sansom[1,3] ✉

The continuous renewal of the skin relies on stem and progenitor cells, yet their differential susceptibility to oncogenic mutations in cutaneous squamous cell carcinoma (cSCC) remains unclear. Rapid cSCC develops in melanoma patients on BRAF inhibitors due to paradoxical MAPK activation. To model this in mice, we use two complementary approaches: HRAS$^{G12V}$ with a BRAF inhibitor to mimic paradoxical MAPK activation, and BRAF$^{V600E}$, which drives MAPK hyperactivation without further treatment. We target these mutations to the interfollicular stem and differentiation-committed progenitors of the basal epidermis. While stem cells rapidly form tumours, progenitors exhibit long-latency resistance despite retaining mutations and repopulating the basal layer. Ultimately, both populations produce similar tumours, showing a shared transformation process. However, SOX2 is uniquely upregulated in progenitor-derived tumours and is expressed in 20% of human cSCC, indicating it might mark tumours arising from committed progenitors. Here, we show that SOX2 overexpression, along with MAPK activation, in progenitors induces a stem-like state and renders this otherwise resistant population permissive to rapid transformation.

Adult stem and progenitor cells located in the basal layer continuously regenerate to maintain the mammalian interfollicular epidermis, ensuring its structural integrity and protection against the external environment. Within this compartment, a stem cell population expressing K14/K5+ coexists with a differentiation-committed progenitor population that additionally expresses differentiation markers, such as IVL+. Two models have been proposed to explain the basal layer heterogeneity. One describes the existence of two distinct basal progenitors that contribute differentially to epidermal homoeostasis, wound healing[1,2] and mechanical stretching[3]. The other proposes that the basal layer heterogeneity arises from a continuum of cells at different levels of commitment towards differentiation in response to cues from the surrounding environment[4]. These cell populations also differ in their ability to initiate basal cell carcinoma[5,6], but whether they exhibit similar functional divergence in cutaneous squamous cell carcinoma (cSCC) remains unclear.

Spontaneous human cSCC exhibits high mutational burden, heterogeneity and copy number alterations accumulated with age and UV exposure[7], including mutations in known drivers of cSCC[8]. The skin hierarchy and spatial compartmentalisation typically restrict the proliferation of mutated cells. However external stressors and promoter events, such as wounding, inflammation, UV exposure and certain oncogenic mutations, can overcome these constraints, leading to tumorigenesis[3,9–13]. The most frequently observed mutations in cSCC accumulate in Notch (79.5%) and TP53 (71%) signalling pathways[10]. Single mutations within these pathways alone are insufficient to induce

[1]Cancer Research UK Scotland Institute, Glasgow, UK. [2]Cancer Research UK Manchester Institute, Manchester, UK. [3]School of Cancer Sciences, University of Glasgow, Glasgow, UK. [4]Present address: Oncodrug Ltd, Macclesfield, UK. [5]Present address: Cancer Research UK Scotland Centre, Institute of Genetics and Cancer, University of Edinburgh, Edinburgh, UK. ✉e-mail: P.Centeno@crukscotlandinstitute.ac.uk; owen.sansom@glasgow.ac.uk

cSCC in mice and additional genetic alterations involving the TGFβ and MAPK signalling pathways are typically required[7,14–16]. Murine models have demonstrated the tumour's ability to arise from different epidermal compartments, including the hair follicle, the sebaceous glands and the interfollicular epidermis[9,14,15,17]. The complexity and slow tumour progression of these models have limited mechanistic insights into skin carcinogenesis.

The most widely used model for cSCC is the well-established two-stage chemical carcinogen 7,12-dimethylbenz[a]anthracene (DMBA), followed by the promoting agent 12-O-tetradecanoylphorbol-13-acetate (TPA)[9,18]. This leads to rapid tumorigenesis from the *Lgr6+* hair follicle stem cells, predominantly driven by *Hras*[Q61L] and *Trp53* mutations[9,19]. While *HRAS* mutations are relatively uncommon in spontaneous human cSCC (9–16%), they are frequent in healthy skin as passenger mutations[8,14,20]. Besides, a rapid onset of HRAS-driven cSCC is commonly found in ~60% of melanoma patients treated with BRAF[V600E] inhibitors (BRAFi)[20–22]. This is attributed to the paradoxical activation of the MAPK signalling pathway driven by BRAFi, which promotes BRAF-CRAF dimers to synergise with pre-existing *HRAS* mutations in the skin[22,23].

Here, we assess the mechanisms that confer permissiveness to initiate cSCC in the interfollicular stem and progenitor basal populations. We develop a suite of genetically engineered mouse models to study the aetiology of rapidly growing cSCC driven by hyperactivation of the MAPK signalling pathway. We define a tumour-primed and a tumour-resistant population and uncover a shared transcriptional programme responsible for transformation. Our findings demonstrate the different susceptibility of basal populations to cSCC transformation and reveal that SOX2 renders the otherwise tumour-resistant progenitor population susceptible to oncogenic transformation.

## Results

### Paradoxical MAPK signalling activation is a tumour promoter in cSCC

To study the susceptibility of the two different basal populations to transformation and tumour formation, we set out to generate genetically engineered mouse models of cSCC that resembled human disease. We drove the expression of *Hras*[G12V] in the basal stem and progenitor populations by using tamoxifen-inducible Cre recombinase expressed under the control of the *Krt14* promoter (*Krt14*-CreERT2; hereafter K14) or the *Ivl* promoter (*Ivl*-CreERT; hereafter Ivl), respectively (Fig. 1a).

Expression of one copy of the *Hras*[G12V] allele (hereafter HRAS[G12V]) in the K14+ or the IVL+ population did not result in tumorigenesis (Fig. 1b, c). This result is consistent with human studies demonstrating that normal-looking skin carries multiple genetic mutations, including oncogenes, and that a promotion event is required to trigger transformation[8,9]. Therefore, we applied topical TPA, a well-known agent that promotes inflammation and stimulates epidermal proliferation, to our models three times a week. However this treatment was also insufficient for transformation and only basal MAPK signalling activation was seen in epidermis carrying one copy of HRAS[G12V], either with or without TPA treatment (Fig. 1d). Two copies of the *HRAS*[G12V] were sufficient to drive higher levels of MAPK activation in both models (Fig. 1d). However when expressed in the K14+ population, mice became unwell, likely due to the essential role of RAS proteins in the oesophagus and forestomach gastrointestinal epithelium before any skin-related malignancies developed. Nevertheless, K14:HRAS[G12V/G12V] mice showed hyperproliferation of the skin basal layer, as evidenced by H&E and increased Ki67 expression (Supplementary Fig. 1a). In the IVL+ basal population, two copies of the allele drove tumorigenesis with a tumour-free survival of 24 days (Fig. 1c and Supplementary Fig. 1a).

Interestingly, melanoma patients treated with BRAFi, such as dabrafenib, commonly develop fast-growing cSCC within weeks of treatment, driven by preexisting *Hras* mutations and paradoxical activation of the MAPK signalling pathway[20–22]. Thus, we treated Ivl:HRAS[G12V/+] and K14:HRAS[G12V/+] mice daily with the BRAFi dabrafenib to mimic the patient dosing regimen. Notably, we confirmed sustained MAPK activation in both models (Fig. 1d) and onset of tumour formation with a tumour-free survival rate of 22 days for the K14:HRAS[G12V/+] model and 58 days for the Ivl:HRAS[G12V/+] model (Fig. 1b, c and Supplementary Fig. 1b). Hereafter, the tumour-free survival rate refers to the time from oncogene induction to the first macroscopically change observed in the skin. BRAFi-promoted tumours exhibited the classic cSCC hallmarks, including keratin pearls, parakeratosis and nuclear dysplasia, together with increased proliferation (Fig. 1e). Moreover we also treated non-tumour-bearing mice carrying one copy of the HRAS[G12V] in the K14+ or IVL+ populations at $180 \pm 5$ days post-oncogene induction with the BRAFi. Following treatment, these aged-treated mice quickly developed tumours within days, which highlights the strong tumour-promoting effect of BRAFi in *Hras* mutant skin and models the development of cSCC in melanoma patients with *Hras* mutations (Fig. 1f).

Overall, we have shown that MAPK activation drives cSCC from stem and committed progenitor epidermal populations. While one copy of *HRAS*[G12V], combined with TPA, causes only basal MAPK activation, two copies of the mutated gene are needed to achieve sufficient activation for transformation. In contrast, BRAFi acts as a potent tumour promoter able to drive MAPK paradoxical activation from a single copy of HRAS-mutated skin (Fig. 1g).

### Tumour-primed and tumour-resistant populations coexist in the skin

To gain further insights into the tumorigenic capacity of these distinct pools of progenitors, we used the potent oncogenic mutation BRAF[V600E/+] (hereafter BRAF[V600E]), which encodes a different downstream oncogenic effector of the pro-proliferative MAPK signalling pathway (Fig. 2a). We generated a suite of mouse models expressing BRAF[V600E] under the control of K14 and *Krt5* (K5-CreERT; hereafter K5) promoters, targeting the stem-cell basal population and the Ivl promoter, targeting the committed basal and suprabasal populations (Fig. 2b and Supplementary Fig. 2a).

Upon BRAF[V600E] induction in the stem-cell population (K5/K14+), cells rapidly activated the MAPK signalling pathway without the need for further promotion (Supplementary Fig. 2b), resulting in fast-growing tumours (9 and 10 days to tumour onset) (Fig. 2c, d). Reducing the dose and duration of the tamoxifen-induction regimen in the K5:BRAF[V600E] and K14:BRAF[V600E] models delayed tumour onset, but tumours still grew rapidly (Supplementary Fig. 2c). Transplantation of subcutaneous pieces from K5:BRAF[V600E]-derived tumours into C57BL/6J and NSG-II2 immunodeficient mice revealed cancer stem cell potential and an ability to form secondary tumours (Supplementary Fig. 2d). Thus, this population is permissive to tumour development and primed for oncogenic transformation. Therefore, we termed it tumour-primed.

In contrast, despite BRAF[V600E]-driven activation of MAPK signalling reaching similar levels in the progenitor population (IVL+) (Supplementary Fig. 2e), tumours exhibited a longer onset latency (125 days to tumour onset) and grew slowly (Fig. 2c, e). This was consistent with tumours showing an increased number of apoptotic regions, marked by caspase-3 (CASP3) and cPARP cleavage, compared to tumours derived from the K14/K5+ population (Supplementary Fig. 2f). The longer latency and increased experimental time allowed this model to develop multiple lesions at various sites, including the back, paws, ears and lips, while the stem-cell models developed only one tumour on their back at the precise area of tamoxifen application at clinical end point (Fig. 2f). The longer latency of tumours arising from the

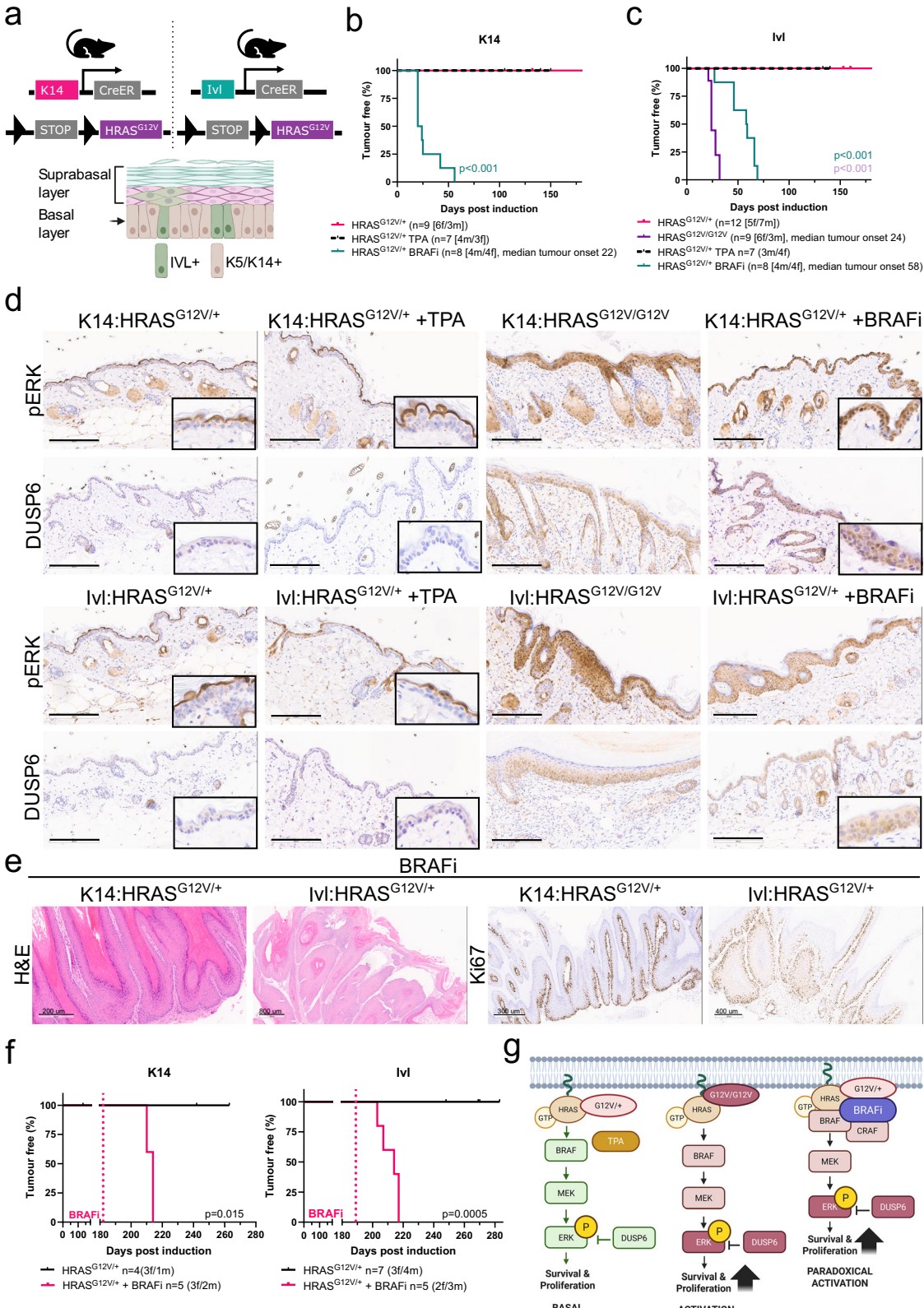

**e**

BRAFi

**f**

**g**

progenitor population suggests that this population requires additional cooperating events to facilitate tumour onset, so we termed this population tumour-resistant.

To promote cSCC in the IVL+ tumour-resistant population, we exposed the back of the Ivl:BRAF[V600E] mice to mild UV [four weekly doses, six standard erythema doses (SED)]. This did not accelerate tumour onset nor increase tumour burden, despite UV being a risk

factor for cSCC (Fig. 2c, f). Similarly, topical treatment with the inflammation-promoting agent TPA (three times per week) did not accelerate tumorigenesis (Supplementary Fig. 2g). No statistical differences in tumour onset were seen between K14:BRAF[V600E] cohorts housed in two different establishments or by sex (Supplementary Fig. 2h, i). However the Ivl:BRAF[V600E] model showed a slightly shorter tumour onset in the cohort housed at the CRUK SI compared to CRUK

**Fig. 1 | Paradoxical MAPK signalling activation is a tumour promoter in cSCC.**
**a** Schematic representation of the mouse models used (top) and the epidermis (bottom) formed by the suprabasal layer, made by IVL+ cells and the basal layer, which includes a mixture of K14+/K5+ and IVL+ cells. Created in BioRender. Centeno, P. (2025) https://BioRender.com/zzh4ohx. Kaplan–Meier tumour-free survival [time of the first lesion or macroscopic change in the skin] plot for K14:HRAS$^{G12V/+}$ (n = 9) and those treated with the BRAFi dabrafenib (n = 8) and TPA (n = 7) (**b**) and Ivl:HRAS$^{G12V/+}$ (n = 12), Ivl:HRAS$^{G12V/G12V}$ (n = 9) and those treated with BRAFi dabrafenib (n = 8) and TPA (n = 7) (**c**). P-values were determined using the log-rank (Mantel–Cox) test and HRAS$^{G12V/+}$ are used as controls. ***p < 0.001.
**d** Representative histological images showing downstream MAPK signalling activation via pERK and DUSP6 IHC in normal skin carrying HRAS$^{G12V}$ mutations, untreated and TPA and BRAFi treated, collected at the end of the experiment. Images representative of four animals per genotype. Scale bar is 200 μm.
**e** Representative histological images, including H&E (scale bar 200 and 800 μm) and Ki67 (scale bar 300 and 400 μm) IHC in tumours derived from K14:HRAS$^{G12V/+}$ and Ivl:HRAS$^{G12V/+}$ mice treated with the BRAFi dabrafenib at the clinical endpoint.

Images representative of four animals per genotype. **f** Kaplan–Meier tumour-free survival plots for K14:HRAS$^{G12V/+}$ (n = 4) and those treated with BRAFi dabrafenib (n = 5) at 182 days after oncogene induction (left) and Ivl:HRAS$^{G12V/+}$ (n = 7) and those treated with BRAFi dabrafenib (n = 5) at 189 days after oncogene induction (right). P-values were determined using the log-rank (Mantel–Cox) test and untreated HRAS$^{G12V/+}$ are used as controls. **g** Schematic representation of the different levels of MAPK signalling pathway activation observed in the skin of HRAS$^{G12V}$ mice, with heterozygous mutation and TPA treatment, homozygous mutation and heterozygous mutation and BRAFi treatment. A single copy of Hras$^{G12V}$ combined with TPA is insufficient to reach the threshold of MAPK signalling activation in the basal layer for tumour formation and two copies are necessary. In contrast, BRAFi acts as a potent tumour promoter, driving MAPK paradoxical activation from a single copy of HRAS-mutated skin. pERK is at the end of the MAPK signalling cascade and is used throughout the study as a direct readout for activation. DUSP6 is activated through a negative feedback loop in response to increased pERK levels. Created in BioRender. Centeno, P. (2025) https://BioRender.com/zzh4ohx. Source data are provided as a Source data file.

---

MI, while no differences were seen between sexes (Supplementary Fig. 2j, k).

## Tumours share histopathological features

We characterised the histopathological features of tumours originating in the K5:BRAF$^{V600E}$, K14:BRAF$^{V600E}$ and Ivl:BRAF$^{V600E}$ models and found remarkable similarities among them. Tumours were formed by vertical columns of atypical keratinocytes, including hyperchromatic and pleomorphic nuclei (Fig. 2g–i and Supplementary Fig. 3a). Detailed histopathological analysis showed distinctive features of cSCC. These included enlarged interfollicular epidermis and cornified layers with an aberrant accumulation of extracellular keratin (keratin pearls), nuclear atypia and incomplete maturation of keratinocytes (parakeratosis) as the keratinocytes in the cornified layer retain their nuclei (Fig. 2g–i).

Regardless of the cell of origin, tumours presented dyskeratosis and abnormal expression of differentiation markers. They show an increase in the number of layers expressing the basal markers KRT5 and KRT14, as well as the suprabasal markers KRT1 and IVL (Supplementary Fig. 3b–d). In the most advanced cases, all markers were expressed throughout the entire epidermal layer, indicating a complete loss of cellular hierarchy and disruption of multi-layer organisation. Moreover the increase in Ki67 expression denotes epidermal hyperproliferation and hyperplasia of the interfollicular epidermis (Supplementary Fig. 3e). These lesions were highly proliferative, recapitulating the rapid onset and histopathology observed in human cSCC.

Overall, despite both populations residing in the basal layer and sharing the microenvironment, they show profound differences in susceptibility to transformation upon oncogene induction. Notably, BRAF$^{V600E}$ mimics the MAPK hyperactivation seen in cSCC from patients treated with BRAFi and harbouring *Hras* mutations in normal-looking skin, making it a valuable model for studying the disease.

## The IVL+ progenitor population is tumour-resistant

In homoeostasis, IVL+ marks the suprabasal layer and a minority population in the basal layer that is committed to differentiation, although its differentiation does not prevent cell cycle entry[1,4]. We aimed to understand how this population resists tumorigenic transformation after oncogene expression.

We followed IVL+ and KRT14+ expressing cells by co-expressing the *Rosa26* LSL-tdTomato fluorescent protein (hereafter tdRFP) reporter in the presence and absence of BRAF$^{V600E}$ (Fig. 3a). We validated the fidelity of our lineage tracing models by co-immunofluorescence imaging of RFP and KRT14, followed by confocal microscopy analysis. In the absence of oncogene activation, the K14:tdRFP model labelled all basal layer cells (Supplementary Fig. 4a), whereas the Ivl:tdRFP model labelled only a few basal cells but most of

the suprabasal cells (Fig. 3b). However when BRAF$^{V600E}$ was expressed in the IVL+ population (Ivl:BRAF$^{V600E}$-tdRFP), we observed a progressive expansion of this population in the basal layer, with RFP and IVL proteins colocalising, further validating the model's fidelity (Fig. 3c).

We then performed lineage-tracing experiments at various time points to compare the RFP-labelled populations between control and oncogene-expressing skin in K14 and Ivl-driven models. These time points, marked by tdRFP and oncogene induction, span from before visible skin changes to day 160 or the clinical endpoint (Fig. 3d).

In the Ivl-tdRFP control model (no oncogene), a ~50% reduction in the label-retaining population in the basal layer was observed from day 8, consistent with committed cell differentiation and delamination (Fig. 3e, f). Besides, suprabasal cells that were not located above an RFP label basal cluster turned over and those regions progressively lost the tdRFP label. After that initial drop, the labelled population remained stable for up to 160 days, marking a long-residing, stable population of the basal layer that was able to self-renew and contribute with its progeny to the suprabasal layers, consistent with previous data[1,4].

In contrast, upon oncogene activation in the Ivl:BRAF$^{V600E}$-tdRFP model, we saw a rapid ~40% increase in the clonal persistence of labelled basal cells, which quickly outcompeted the unlabelled population (Fig. 3e, f). At day 15, the entire basal layer was labelled to an equivalent level to the K14+ labelled population and tumours sampled at end point retained tdRFP (Fig. 3e and Supplementary Fig. 4a). IVL protein IHC staining also confirmed that IVL+ cells became the majority population in the basal layer in the presence of BRAF$^{V600E}$, whilst retaining the expression of this differentiation commitment marker (Supplementary Fig. 4b). Moreover in the Ivl:BRAF$^{V600E}$-tdRFP model, there was an increase in epidermal thickness compared to the control, which decreased over time but remained significant throughout the study (Fig. 3g). Notably despite most basal cells carrying the oncogenic mutation, constraints remained in place, resisting transformation.

In the K14-driven models, the standard induction regimen (4 times over 7 days) labelled all cells in the basal layer and their progeny in the suprabasal layer to saturation (Supplementary Fig. 4c). Therefore, we repeated the study at the lower induction dose regimen (single application), allowing us to study the dynamics of single clones. By day 8, quantification of RFP+ cells revealed an almost two-fold increase in the basal population in the oncogene-carrying group, which translated into rapidly growing tumours by day 15 (Fig. 3h, i). Tumours and hyperplastic lesions also exhibited increased expression of the IVL-suprabasal marker throughout all layers, indicating a previously observed breakdown of the skin hierarchy from early timepoints (Supplementary Fig. 4d).

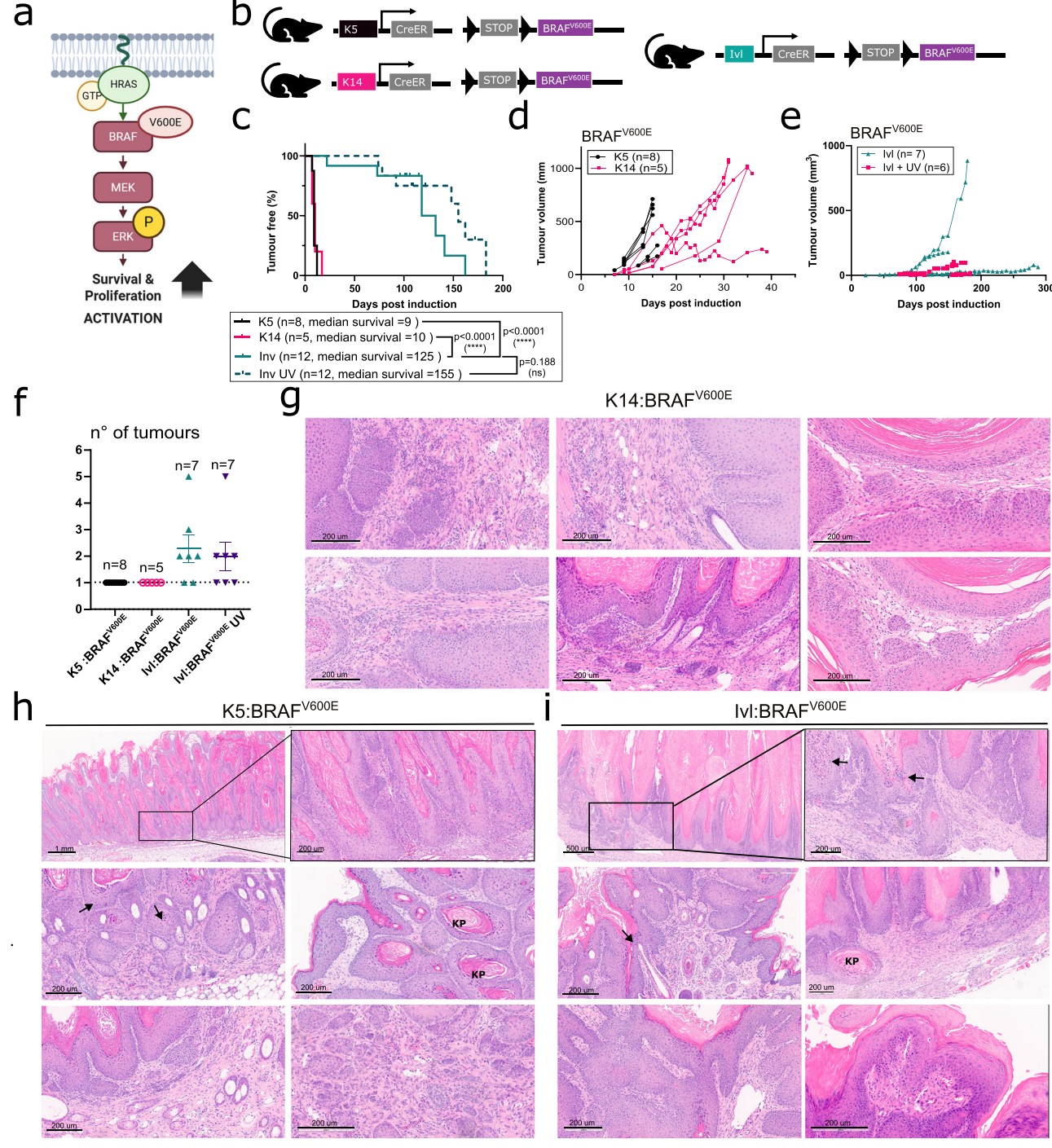

**Fig. 2 | Tumour-primed and tumour-resistant populations coexist in the basal layer. a** Schematic representation of the activation of the MAPK signalling pathway observed in the skin of BRAF[V600E] mice. Created in BioRender. Centeno, P. (2025) https://BioRender.com/zzh4ohx. **b** Schematic representation of the mouse models used for skin tumorigenesis driven from K5, K4 and Ivl promoters. **c** Kaplan–Meier plot displaying tumour-free survival for K5:BRAF[V600E] (n = 8), K14:BRAF[V600E] (n = 5), Ivl:BRAF[V600E] (n = 12) and Ivl:BRAF[V600E] UV-treated (n = 12) female mice aged until clinical endpoint. P-values were determined using the log-rank (Mantel–Cox) test. NS not significant. Total tumour growth in K5:BRAF[V600E]

(n = 8) and K14:BRAF[V600E] (n = 5) (**d**) and in Ivl:BRAF[V600E] (n = 7) and UV-treated Ivl:BRAF[V600E] (n = 6) female mice (**e**). **f** Total tumour burden (mean and SEM) in K5:BRAF[V600E] (n = 8), K14:BRAF[V600E] (n = 5), Ivl:BRAF[V600E] (n = 7) and UV-treated Ivl:BRAF[V600E] (n = 7) female mice at clinical endpoint. Representative H&E of K14:BRAF[V600E] (**g**), K5:BRAF[V600E] (**h**) and Ivl:BRAF[V600E] (**i**) tumours at clinical endpoint showing keratin pearls (KP) and nuclear atypia (arrows). Images representative of five animals per genotype. Scale bar is 200 μm unless otherwise stated in the image. Source data are provided as a Source data file.

Varying levels of MAPK signalling activation could not explain the difference in tumour permissivity observed between these populations, as high levels of pERK were observed in both models from early time points (Supplementary Fig. 4e).

Together, these data demonstrate that oncogene expression in the committed IVL+ population led to a clonal proliferation and expansion similar to the stem-cell K14+ population. However it stays tumour-restrictive for a longer period. This indicates that

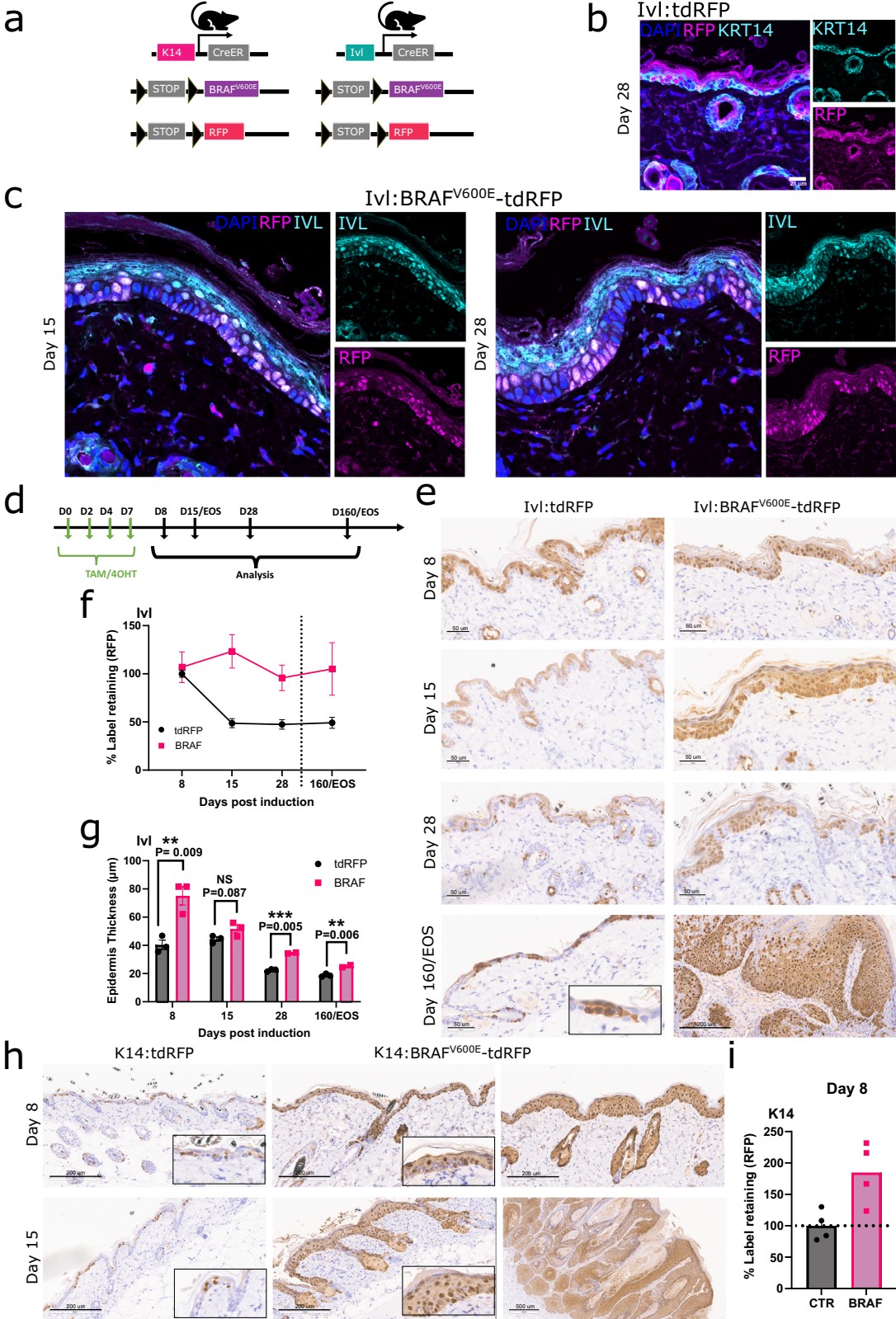

transformation constraints are still active in the IVL+ population, even though oncogene-bearing cells have colonised the basal layer.

## cSCC shared transcriptional profile

Given the histological similarities, we next conducted transcriptional profiling of the total tumour tissue to identify potential differences and vulnerabilities in these tumours. After batch correction

(Supplementary Fig. 5a), we merged the K5:BRAF$^{V600E}$ and K14:BRAF$^{V600E}$ tumour samples as they clustered independently of the model and our analysis returned no differentially expressed genes (Supplementary Fig. 5b, c). Note that these tumours arose from the same cell populations simultaneously expressing KRT5 and KRT14. Then, we compared K14/K5:BRAF$^{V600E}$- and Ivl:BRAF$^{V600E}$-derived tumours against normal skin from control littermates. Unsupervised

**Fig. 3 | Oncogene-expressing IVL+ tumour-resistant population takes over the entire skin but remains tumour-restrictive. a** Schematic representation of the mouse models used for lineage tracing of the KRT14+ and IVL+ populations. **b, c** Representative Z-stack co-immunofluorescence (IF) of KRT14 and RFP of skin derived from Ivl:tdRFP (control) at day 28 post-induction (**b** Scale bar is 21 μm) and IVL and RFP of skin derived from Ivl:BRAF$^{V600E}$-tdRFP at day 15 and 28 post-induction (**c** images taken on a 50× objective). Images representative of three animals per genotype. **d** Schematic representation of the experimental design used for lineage tracing. **e** Representative RFP IHC at different time points post-induction in Ivl:tdRFP and Ivl:BRAF$^{V600E}$-tdRFP. Scale bar is 50 μm unless otherwise stated in the image. Images representative of three animals per genotype and time point. **f** Quantification of the clonal persistence (%RFP) in the basal epidermis region of the skin from Ivl:tdRFP and Ivl:BRAF$^{V600E}$-tdRFP mice at different times post-induction. Individual measurements averaged from three different skin stripes per animal. N = 3 independent mice per timepoint in each arm. Data are presented as mean values +/− SEM. **g** Epidermis thickness quantification at different time points post-induction in Ivl:tdRFP and Ivl:BRAF$^{V600E}$-tdRFP. Ten measurements from three different skin stripes per animal. Points represent individual animals, three animals per time point. Data are presented as mean values +/− SEM. Unpaired two-tail t-test. NS not significant P > 0.05, **p < 0.01 and ***p < 0.001. **h** Representative RFP IHC at different time points post-induction in K14:tdRFP and K14:BRAF$^{V600E}$-tdRFP skin and tumours. Data representative of two independent animals and three skin regions per group and time point. Scale bar is 200 μm unless otherwise stated in the image. **i** Quantification of the clonal persistence (%RFP) in the basal epidermis region of the skin from K14:tdRFP and 14:BRAF$^{V600E}$-tdRFP mice at day 8 post-induction. N = 4 from 2 independent mice per timepoint in each arm. Source data are provided as a Source data file.

clustering and principal component analysis (PCA) showed that the tumours separated from the normal skin samples (Fig. 4a).

Both tumour groups revealed a large number of transcriptionally upregulated genes when compared to normal skin (Fig. 4b, c). Notably, most upregulated genes (82.8%) were shared between the K14/K5:BRAF$^{V600E}$ and Ivl:BRAF$^{V600E}$ tumours (Fig. 4d), of which 253 were expressed significantly higher in tumours compared to normal skin (log fold change >3 and padj <0.001; Supplementary Data 1). Digital sorting of the bulk transcriptome and deconvolution into previously defined skin populations[24], revealed that most cells (~70%) were keratinocytes belonging to the permanent part of the epidermis, with some samples also containing keratinocytes derived from hair follicles (Supplementary Fig. 5d). Fibroblasts made up to 10% of the total transcriptome analysed and overall tumour composition across K14, K15, or Ivl-derived tumours was similar.

Gene set enrichment analysis highlighted commonly upregulated signatures, including those associated with the *Myc*, *Lgr5* and *Lgr6* wound healing response[25], embryonic epidermis[26] and the human-derived cSCC tumour-specific keratinocyte population[27] among others (Fig. 4e and Supplementary Fig. 5e). Importantly, no major differences were seen in pathway enrichment between models. It is possible that these results may reflect variations in epithelial/stroma ratios. Therefore, to validate the transcriptional findings in the epithelial cells, we assessed protein levels using IHC directly in the tissue.

We examined the expression of individually selected differentially expressed genes that were enriched in the tumours compared to normal skin, including *Myc*, the cSCC markers *Krt6a* and *Krt6b*[28], along with *Ly6a and Anxa1* ligand (Fig. 4f). ANXA1 is normally secreted by wounded epithelia during tissue regeneration[25] and is enriched in the tumour-specific keratinocyte signature[27]. Interestingly, a model for tissue injury-sensing uncovered the role of IL24, produced at wound sites in response to hypoxia, in promoting epithelial proliferation and re-epithelialization via pSTAT3 activation[29]. This pathway may also extend to cSCC, as evidenced by increased expression of the hypoxia sensor HIF1α, IL24 and pSTAT3 in our models (Fig. 4f, g). We also observed elevated *Il6*, which stimulates the release of proinflammatory cytokines from the cutaneous microenvironment in response to wounding, activating TGFβ receptor signalling and the STAT3 signal transduction pathway[29,30].

These transcriptomic results were validated by RNA *in situ hybridisation* for *Ly6a* and IHC for MYC, ANXA1 and pSTAT3 in our models driven by BRAF$^{V600E}$ and HRAS$^{G12V}$ in combination with BRAFi (Fig. 4g, h and Supplementary Fig. 6a). Similar findings were seen in tumours originating from the hair follicle stem cells upon MAPK signalling hyperactivation and deletion of the TGFβ receptor 1 (*Alk5*) (Lgr5:BRAF$^{V600E}$-ALK5$^{fl/fl}$, Lgr5:HRAS$^{G12V/G12V}$-ALK5$^{fl/fl}$, Lgr5:KRAS$^{G12D/+}$-ALK5$^{fl/fl}$)[14] and deletion of *Trp53* and *Notch2* (Lgr5:TRP53$^{fl/fl}$-NOTCH2$^{fl/fl}$) (Supplementary Fig. 6b).

Interestingly, oncogene expression in the tumour-primed population (K14-BRAF$^{V600E}$-tdRFP) exhibited strong pSTAT3 and MYC

activation from day 8 in hyperplastic lesions, which persisted later in the tumours (Supplementary Fig. 6c). In contrast, oncogene expression in the tumour-resistant population (Ivl-BRAF$^{V600E}$-tdRFP) exhibited high pSTAT3 levels at early time points, but these diminished and became limited to small clones by day 28, with only weak MYC activation observed across the time course (Supplementary Fig. 6c).

Overall, these results reveal shared molecular and transcriptional hallmarks of epidermal transformation that rely on MYC and pSTAT3 activation for transformation and are independent of the cell of origin, tumour latency, or oncogenic driver mutations.

**SOX2 is expressed in tumours derived from the IVL+ population**

We next assessed whether transcriptional profiling could identify potential markers to distinguish tumours arising from the different cell populations. Thus, we compared the transcriptomic profile of tumours from K5/K14:BRAF$^{V600E}$ and Ivl:BRAF$^{V600E}$ model, which revealed only a handful of differentially expressed genes (Fig. 5a). Notably, SOX2 emerged as a key transcriptional difference, being highly enriched in Ivl:BRAF$^{V600E}$ tumours, relative to K5/K14:BRAF$^{V600E}$ tumours. SOX2 is a super pioneer transcription factor capable of binding to closed chromatin through its ability to interfere with the maintenance of DNA methylation[31]. Furthermore, SOX2 is commonly used as a marker of cancer stem cells and is upregulated in ~20% of cSCC patients, where it rewires cells for tumour initiation and growth[32,33].

We confirmed SOX2 upregulation in Ivl:BRAF$^{V600E}$ tumours compared to K5:BRAF$^{V600E}$ or K14:BRAF$^{V600E}$ tumours by IHC (Fig. 5b). Similar results were obtained in HRAS-driven tumours, where SOX2 expression was only seen in those arising from the IVL+ population (Fig. 5c). SOX2 expression was negligible in normal skin during homoeostasis or in TPA-treated skin, suggesting that its expression is not linked to inflammation (Fig. 5d). SOX2 in combination with CD34 have been proposed to mark 'tumour initiating cells' in cSCC[33]. Thus, we stained K14 and Ivl-derived tumours taken at clinical endpoint for CD34 but found no expression (Supplementary Fig. 7a). We did not detect SOX2 expression or CD34 in either of the models at earlier time points, while the skin remained histologically normal (Supplementary Fig. 7b, c).

We extended our investigation to a broader suite of models. We found SOX2 expressed in DMBA/TPA-induced tumours, as previously shown[33], but not in tumours originating from *Lgr5*+ cells (hair follicle bulge) driven by different oncogene combinations (Supplementary Fig. 7d, e). We also assessed the expression of SOX9, a distinct regulator of chromatin accessibility that controls hair follicle stem cell fate and is activated during tumorigenesis[34]. In contrast to SOX2, SOX9 show not cell specificity, as it expression expanded from hair follicles in normal skin to the basal layer in tumours across all models, regardless of cell of origin (Supplementary Fig. 8a). The suprabasal markers KRT4 and KRT13, commonly expressed in skin homoeostasis, were also differentially upregulated at RNA level in the tumours derived from the Ivl:BRAF$^{V600E}$ model (Fig. 5a). However we found no

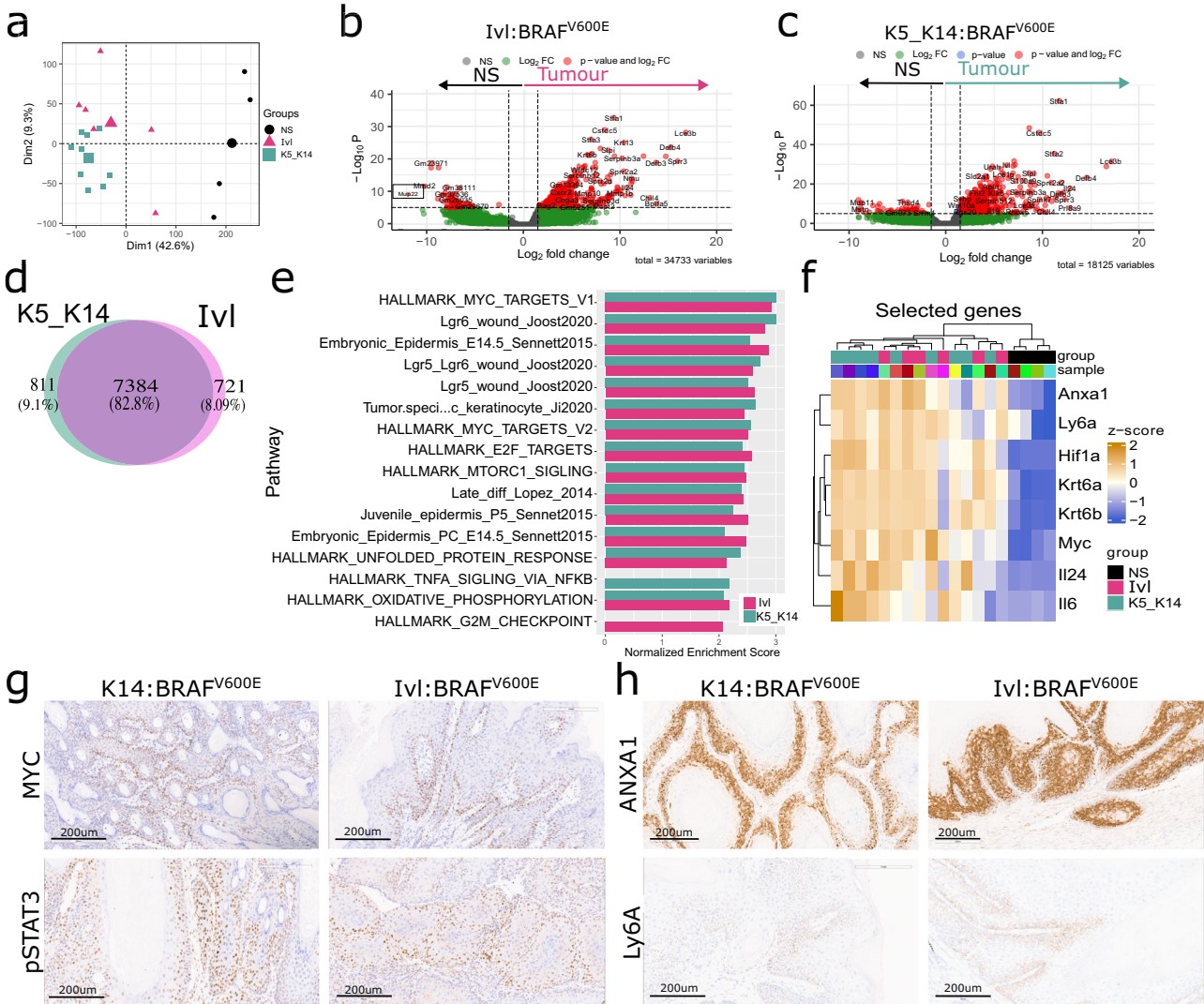

**Fig. 4 | Shared transcriptional profile independent of the cell-of-origin or oncogene. a** Principal component analysis (PCA) of normalised expression values showing the correlation between the transcriptional profiles of normal skin (NS, n = 4), Ivl:BRAF^V600E (Ivl, n = 6) and K5/K14:BRAF^V600E (K5_K14, n = 9). Volcano plots showing differentially expressed genes in Ivl tumours (n = 6) (**b**) and K5_K14 tumours (n = 9) (**c**) compared with normal skin (n = 4) using the Wald test (two-tailed). **d** Venn diagram showing upregulated genes shared between Ivl:BRAF^V600E (n = 6; Ivl) and K5/K14:BRAF^V600E (n = 9; K5_K14) tumours vs normal skin (n = 4). **e** Gene set enrichment analysis of Hallmarks and indicated pathways in K5_K14

(n = 9) tumours and Ivl (n = 6) tumours. Showing pathways significantly enriched (padj <0.01 and NES >2 based on an adaptive multi-level split Monte–Carlo). **f** Hierarchical clustering heatmap of selected genes showing normalised expression and correlation between normal skin (NS; n = 4), Ivl:BRAF^V600E (Ivl; n = 6) and K5/K14:BRAF^V600E (K5_K14; n = 9). **g, h** Representative images of IHC validation of selected targets in K14:BRAF^V600E and Ivl:BRAF^V600E, including MYC, pSTAT3, Ly6A and ANXA1 at clinical endpoint. Images representative of four animals per geno-type. Scale bar is 200 μm.

substantial differences at protein level across models (Supplementary Fig. 8b).

To investigate how these data translate to human cSCC, we reanalysed a human transcriptional dataset with different stages of disease progression[7]. We observed that *SOX2* levels were increased in 27% of premalignant actinic keratosis (AK) and 35% of cSCC lesions, whereas *SOX2* expression levels were low in normal skin (NS) (Fig. 5e, f). SOX2 levels did not correlate with the patient's age or tumour aggressiveness (tumour depth or diameter) (Supplementary Fig. 8c). We then examined SOX2 protein expression in a tissue microarray containing 250 human cSCC[35], which ranged from no expression, in most of the samples, to mid and high in 21.7% of the histocores (Fig. 5g). These results are consistent with previous studies that reported that ~25% of human cSCC express high or medium levels of SOX2[32,33]. These data underscore a diverse cell origin for cSCC, where SOX2 could mark a subset of

tumours emerging from the IVL+ committed tumour-resistant population.

**SOX2 renders the IVL+ population susceptible to tumorigenesis**

Given the potential role of SOX2 activation in skin tumorigenesis, we assessed whether it is required for cSCC development in a cell-specific manner. We crossed the K14:BRAF^V600E and Ivl:BRAF^V600E models with mice harbouring a conditional *Sox2* knockout allele (hereafter SOX2^fl) (Fig. 6a). Whilst conditional *Sox2* deletion did not affect tumorigenesis or tumour growth in the K14:BRAF^V600E·SOX2^fl/fl model, it significantly delayed tumour onset and tumour growth in the Ivl:BRAF^V600E·SOX2^fl/fl model (Fig. 6b–e). Loss of *Sox2* on its own had no impact on skin homoeostasis in either population, consistent with its lack of expression in normal skin.

The histopathological features of K14:BRAF^V600E·SOX2^fl/fl and Ivl:BRAF^V600E·SOX2^fl/fl tumours were similar to those expressing the

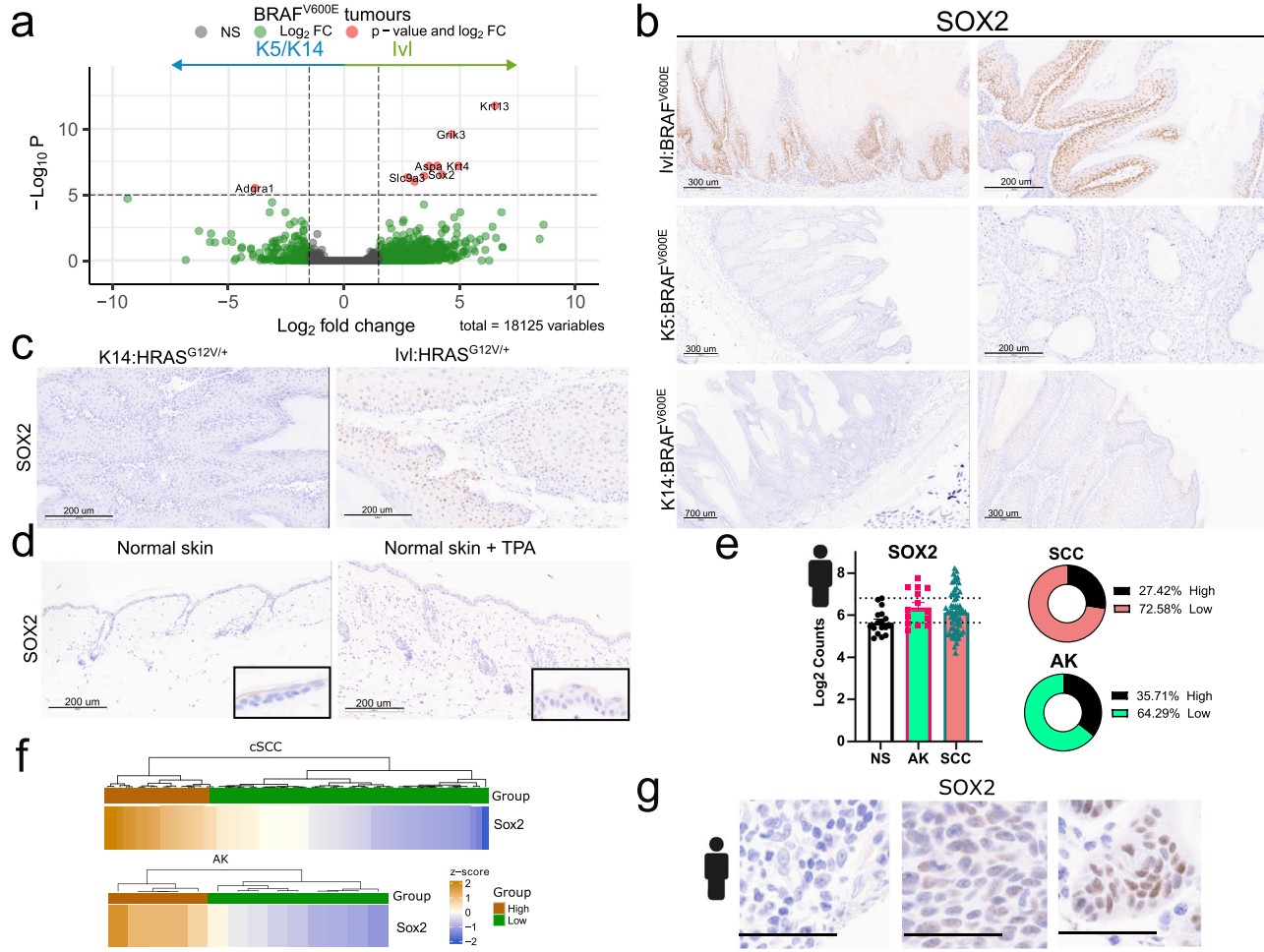

**Fig. 5 | SOX2 is specifically expressed in tumours derived from the IVL+tumour-resistant population. a** Volcano plot showing differentially expressed genes between K5/K14:BRAF$^{V600E}$ (n = 9) and Ivl:BRAF$^{V600E}$ (n = 6) tumours using Wald test (two-tailed). Representative images of IHC validation of SOX2 in Ivl:BRAF$^{V600E}$, K5:BRAF$^{V600E}$ and K14:BRAF$^{V600E}$ (**b**) and BRAFi treated K14:HRAS$^{G12D/+}$ and Ivl:HRAS$^{G12D/+}$ tumours (**c**) at clinical endpoint. Images representative of four animals per genotype. Scale bars are 200, 300 and 700 μm. **d** Representative images of IHC of SOX2 in normal skin (left) and normal skin treated with TPA (right). Images representative of three animals per genotype. Scale bar is 200 μm. **e** Normalised RNA expression levels (log$_2$ counts) of SOX2 from a cSCC human dataset[7], including normal skin (NS, n = 17) and different levels of disease

progression [actinic keratosis (AK, n = 14) and cutaneous squamous cell carcinoma (SCC, n = 66)]. Dotted lines mark the mean *SOX2* normalised expression in normal skin and two standard deviations from the mean. Pie charts showing the percentage of samples with high and low *SOX2* normalised expression for cSCC and AK groups. Two standard deviations from the normal skin mean were used as a threshold. **f** Heatmap showing individual *SOX2* normalised expression levels range in AK (n = 5 high vs n = 9 low) and cSCC (n = 32 high vs n = 30 low) groups from a human dataset[7]. **g** Representative IHC of SOX2 conducted in a human tissue microarray containing 250 cSCC histocores described in ref. 35. 196 (78.4%) of the samples were negative and 54 (21.6%) were positive for SOX2 protein expression. The scale bar is 50 μm.

wild-type *Sox2* allele, including an enlarged epidermis displaying vertical columns of keratinocytes, incomplete maturation of keratinocytes, increased proliferation and MYC and pSTAT3 activation (Supplementary Fig. 9a). Interestingly, the slow-growing Ivl:BRAF$^{V600E}$-SOX2$^{fl/fl}$ tumours show patches escaping recombination and retaining SOX2 protein expression, as demonstrated by IHC, which highlights its critical role in tumorigenesis in this model (Supplementary Fig. 9b). Expression of SOX2 was not observed in the K14:BRAF$^{V600E}$-SOX2$^{fl/fl}$ tumours (Supplementary Fig. 9b).

Next, we assessed whether the overexpression of SOX2 could render IVL+ cells competent for tumorigenesis when coupled to a BRAF mutation. To this end, we interbred an allele that allows the inducible overexpression of SOX2 from the Rosa26 locus [*Rosa26*LSL-SOX2-IRES-eGFP (hereafter SOX2$^{LSL}$)] into the IVL+ tumour-resistant population. The resulting Ivl:BRAF$^{V600E}$-SOX2$^{LSL}$ model gave rise to fast-growing tumours and significantly accelerated tumour onset from 65 to 31 days (Fig. 6c, f). Meanwhile, SOX2 overexpression alone showed no phenotype (Fig. 6c).

The Ivl:BRAF$^{V600E}$-SOX2$^{LSL}$ tumours exhibited advanced histological features of cSCC, including poorly differentiated epithelial, keratin pearls, parakeratosis and invasion. These tumours showed enhanced and disorganised proliferation beyond the basal layer and activated the previously described shared transcriptional profile, which includes pSTAT3, ANXA1 and MYC, along with high SOX2 expression (Fig. 6g).

Importantly, overexpression of SOX2 in the IVL+ tumour-resistant population overrode the activation of the apoptotic programme, marked by the expression of activated cleaved CASP3 and PARP seen before in the Ivl:BRAF$^{V600E}$ model (Fig. 6g and Supplementary Fig. 2f). Tumours derived from the Ivl:BRAF$^{V600E}$-SOX2$^{LSL}$ model show reduced expression of apoptotic markers, similar to those found in tumours derived from the K14 and K5 models (Fig. 6h). This reduction of apoptosis was accompanied by an increased expression of the stem cell marker CD34, marking tumour initiating cells[33] (Fig. 6i).

To understand the kinetics and mechanisms by which SOX2 promotes tumorigenesis, we sampled skin at different times post-induction in the presence and absence of BRAF$^{V600E}$ (Fig. 6j). In the

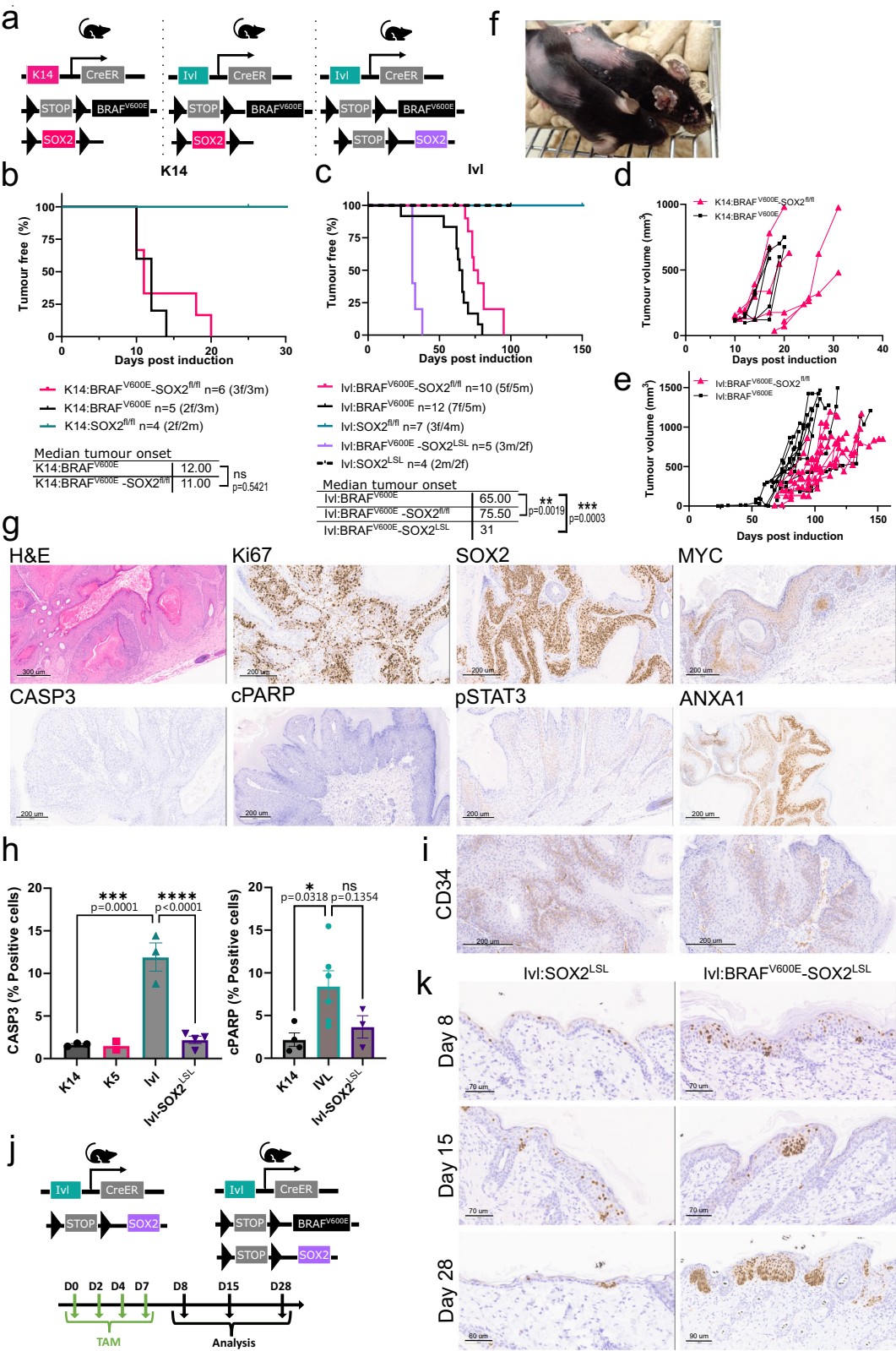

g H&E | Ki67 | SOX2 | MYC

CASP3 | cPARP | pSTAT3 | ANXA1

i CD34

k Ivl:SOX2^LSL | Ivl:BRAF^V600E-SOX2^LSL
Day 8 / Day 15 / Day 28

absence of the oncogene, the Ivl:SOX2^LSL expressed SOX2 mainly in the suprabasal layer with some basal layer cells (Fig. 6k). By day 28, the remaining SOX2-labelled cells were limited to a few clones in the basal layer, as the suprabasal cells overexpressing this protein had undergone turnover. In the presence of the oncogene (Ivl:BRAF^V600E-SOX2^LSL), we initially observed a high number of cells expressing SOX2 in the suprabasal layer at day 8; however from day 15, clones expanding in the basal population became more prominent (Fig. 6k). CD34 expression was also seen from day 15 and maintained in tumours consistent with a population of tumour initiating cells[33] (Supplementary Fig. 9c). These cells did not follow the normal differentiation route towards delamination and remained proliferating and anchored to the basal layer. By day 28, some of the basal clones have expanded and the skin showed dysplastic regions that later progress to tumours. At this

**Fig. 6 | SOX2 overexpression in combination with MAPK activation renders the IVL+ tumour-resistant population permissive to cSCC. a** Schematic representation of the mouse models used for tumorigenesis. Kaplan–Meier tumour-free survival plot for K14:BRAF[V600E] (n = 5), K14:BRAF[V600E]-SOX2[fl/fl] (n = 6) and K14:SOX2[fl/fl] (n = 4) (**b**) and for Ivl:BRAF[V600E] (n = 12), Ivl:BRAF[V600E]-SOX2[fl/fl] (n = 10), Ivl:SOX2[fl/fl] (n = 7), Ivl:BRAF[V600E]-SOX2[LSL] (n = 5) and Ivl:SOX2[LSL] (n = 4) (**c**). P-values were determined using the log-rank (Mantel–Cox) test. NS not significant P > 0.05, **p < 0.01, ***p < 0.001. Note that K14:BRAF[V600E] and Ivl:BRAF[V600E] cohorts correspond to CRUK-SI cohorts also shown in Supplementary Fig. 2 and are shown here for comparison. Total tumour burden growth curves for K14:BRAF[V600E] (n = 5) and K14:BRAF[V600E]-SOX2[fl/fl] (n = 6) (**d**) and for Ivl:BRAF[V600E] (n = 12) and Ivl:BRAF[V600E]-SOX2[fl/fl] (n = 10) (**e**). **f** Representative picture showing Ivl:BRAF[V600E] and Ivl:BRAF[V600E]-SOX2[LSL] littermates at day 30 post oncogene induction. Representative of Ivl:BRAF[V600E] (n = 12) and Ivl:BRAF[V600E]-SOX2[LSL] (n = 5) cohorts. **g** Representative H&E and IHC of Ki67, SOX2, MYC, pSTAT3, CASP3, cPARP and ANXA1 in Ivl:BRAF[V600E]-SOX2[LSL] tumour at clinical endpoint. Images representative of four animals per genotype. Scale bar is 200 μm unless otherwise stated in the image. **h** Apoptosis quantification (% cleaved CASP3 and cPARP of positive cells) in tumours driven by BRAF[V600E] from different populations (CASP3; K14 n = 3, K5 n = 2, Ivl n = 3, Ivl-SOX2[LSL] n = 4 and cPAPR; K14 n = 4, Ivl n = 6, Ivl-SOX2[LSL] n = 4 independent tumours). Quantified area from a total of 15 mm², including an average of 23240.73 cells. Two-way ANOVA multiple comparison. ***p < 0.001 and ****p < 0.0001. Data are presented as mean values +/− SEM. **i** Representative IHC of CD34 in Ivl:BRAF[V600E]-SOX2[LSL] tumour at clinical endpoint. Images representative of four animals per genotype. Scale bar is 200 μm. **j** Schematic representation of the mouse models used for tumorigenesis and experimental design. **k** Representative IHC of SOX2 in Ivl:SOX2[LSL] and Ivl:BRAF[V600E]-SOX2[LSL] at different time points. Images representative of three animals per genotype and per time point. Scale bar is 70 μm unless otherwise stated in the image. Source data are provided as a Source data file.

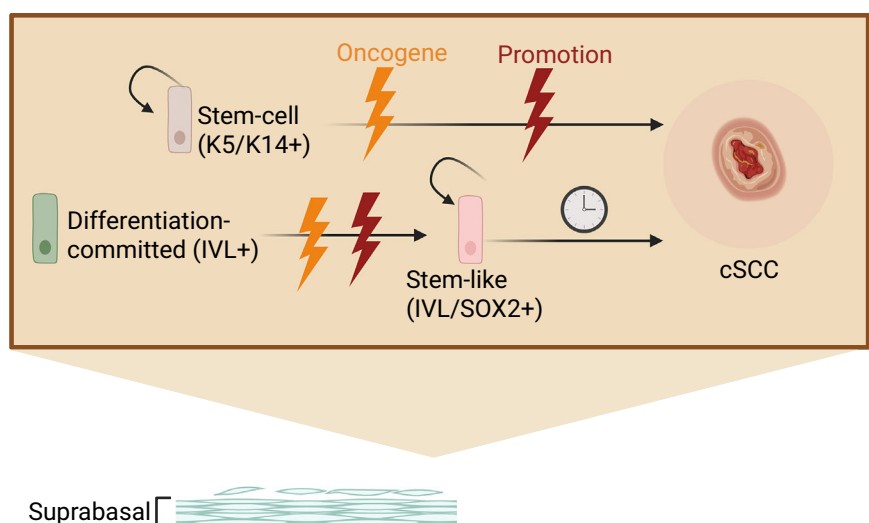

**Fig. 7 | SOX2 confers tumour permissiveness to the IVL+ tumour-resistant population.** Graphic abstract of the model arising from this study. cSCC can be initiated from a stem cell or differentiation-committed progenitor. The latter will require SOX2 activation and will take longer to escape constraints and trigger tumorigenesis. Oncogenic-driven SOX2 activation induces stem-like features critical for tumorigenesis. During this process, actively dividing oncogene-bearing IVL+ progenitors expand through the entire epithelium with little histological change, preparing for tumour initiation and eventually leading to tumorigenesis. We propose a common oncogenic programme, seen in human cSCC[27] and promoted by inflammation, injury or paradoxical activation of MAPK, which allows the transformation of 'normal' oncogene-bearing epithelial cells into a tumour without additional oncogenic events. Created in BioRender. Centeno, P. (2025) https://BioRender.com/zzh4ohx.

time, we could already see BRAF[V600E]-SOX2+ keratinocytes invading the dermis (Fig. 6k). Histological analysis of the skin at different timepoints revealed increased MAPK signalling activation, as evidenced by high levels of DUSP6, along with increased MYC and pSTAT3 expression levels starting from day 8 (Supplementary Fig. 9c).

Transcriptome analysis of bulk tumours derived from Ivl:BRAF[V600E]-SOX2[LSL] showed a large number of differentially expressed genes compared to tumours derived from Ivl:BRAF[V600E] taken at the clinical endpoint (Supplementary Fig. 10a–d). Ivl:BRAF[V600E]-SOX2[LSL] transcriptome further activated pathways such as pSTAT3, epithelial-mesenchymal transition, angiogenesis and immune and inflammation-related responses, among others, to support the faster progression of tumour growth (Supplementary Fig. 10e). From those, we derived a 'SOX2_tumorigenesis' signature with the top 150 differentially expressed genes (log₂ fold change >2 and padj <0.01) of which 139 had direct human orthologs (Supplementary Data 2) and conducted single-sample gene set enrichment in a cSCC human dataset[7]. Consistent with our murine-derived data, human cSCC samples enriched for our 'SOX2_tumorigenesis' signature showed lower enrichment in the apoptosis hallmark (Supplementary Fig. 10f).

Overall, these data suggest that SOX2 is a cell-specific requirement for the IVL+ tumour-resistant progenitor population to initiate cSCC. SOX2 activation in this population, combined with an oncogene, results in a clonal behaviour similar to the K14/K5+ tumour-primed population and promotes the re-acquisition of stemness and tumour-initiating capabilities in the skin's basal compartment. The resulting IVL+ derived tumours shared most histological features and a core transcriptional transformation programme with those originating from a K14/K5+ stem cell population (Fig. 7).

## Discussion

We generated genetically engineered mouse models that recapitulate the rapid cSCC development seen in melanoma patients treated with BRAFi, due to paradoxical MAPK signalling activation[20–22]. High levels

of MAPK signalling activation in the epidermis were critical for transformation. While a single copy of HRAS[G12V] was insufficient, treatment with BRAFi rapidly promoted tumours. Similarly, BRAF[V600E] stimulated MAPK signalling hyperactivation, also promoting tumour development. This data suggests that the skin can tolerate oncogenic mutations with minimal histological changes, but can rapidly progress to tumorigenesis when exposed to a specific stimulus or promoter.

Notably, we obtained rapidly developing tumours within days of oncogene induction in the K14/K5+ basal stem cell population; hence, we called this population tumour-primed. However the IVL+ differentiation-committed progenitor population remained normal despite the oncogene-bearing cells colonising the basal epithelium within a week. Thus, we termed this population tumour-resistant. Oncogene-expressing IVL+ cells remained in the basal layer, where the proliferative environment is confined, challenging their differentiation commitment. This allowed for a longer latency period before tumour development.

Tumours from both basal populations were histologically indistinguishable and showed the features of cSCC, including keratin pearl formation, dermal invasion, nuclear atypia, parakeratosis and loss of skin epidermal hierarchy. They also shared transcriptional programmes enriched for wound-healing[25] and embryonic[26] signatures and for a tumour-specific keratinocyte population seen in human cSCC[27]. In addition, we observed strong MYC activation, which has been proposed as a mechanism to confer stem cell properties and dedifferentiation in the *Gata6*+ epidermal populations[12]. We also identified evidence of hijacking a tissue injury-sensing and wound repair pathway activated by hypoxia and driven by IL24 and pSTAT3[29]. These results highlight the close interplay between proliferation during tumorigenesis and re-epithelialisation during wound healing.

Most of our understanding of cSCC in the mouse has come from seminal work using DMBA/TPA[9,18] or, more recently, mouse models carrying different combinations of oncogenes, including KRAS or other MAPK mutations in combination with TP53[7,14,15]. Despite the diversity of oncogenic events across these models, they also converge on the core set of skin transformation pathways described above and including MYC and hypoxia/STAT3 activation.

One of our key findings was that SOX2 activation in the IVL+ tumour-resistant population is sufficient to render this population susceptible to oncogenic transformation. SOX2 overexpression in combination with oncogene expression rapidly accelerated tumorigenesis in this population in vivo and reduced apoptosis. Moreover these tumours expressed the stem-cell marker CD34 which together with SOX2 have been proposed to mark a population of 'tumour initiating cells'[33]. Conversely, loss of SOX2 in the IVL+ tumour-resistant population further prevented tumour onset and growth upon oncogene expression, while it did not affect the K5/K14+ tumour-primed population.

This SOX2-driven stemness state also appears to be activated in tumours derived from the LGR6+ population in the DMBA/TPA model[9]. Indeed, ablation of SOX2 in the K14+ lineage delayed tumour growth in this model. Since the LGR6+ hair follicle population also expresses K14, we hypothesise that this LGR6+/K14+ population requires SOX2 for transformation, similar to the IVL+ population, which also expresses K14+ (IVL+/K14+). Meanwhile, the interfollicular stem cell basal K14+ (IVL-/LGR6-) population would not rely on SOX2-driven stemness for transformation.

Although the role of SOX2 in cSCC initiation has been previously described[32,33], our findings clarify that its requirement is cell-specific. As a pioneer factor, SOX2 can bind closed chromatin and rewire the epigenome to drive stem cell features[31,36]. In the context of SCC, a bistable transcriptional network, involving SOX2, TRP63 and PITX1, has been proposed to promote cell proliferation and to inhibit differentiation by acting on KLF4[37]. This network may enable tumour cells to switch between proliferation and self-renewal or differentiation and

keratin pearl formation. Building on this model, we propose that SOX2 expression enables IVL+ progenitors to escape their differentiation programme and adopt a more plastic, transformation-permissive state. This is consistent with the recent model of fluid and gradual cell differentiation[4], in which committed cells residing in the basal layer have not necessarily exited the cell cycle and thus remain susceptible to reprogramming and oncogenic transformation.

Importantly, SOX2 activation occurs in ~20% of cSCC patients[32,33]. This suggests that a differentiation-committed cell that has regained stemness potential, such as basal IVL+ or follicular LGR6+, could be the cell of origin in these tumours. Unfortunately, although there is genetic data available for cSCC from melanoma patients treated with BRAFi and these have high levels of HRAS mutations[22], there is no transcriptomic data available to explore whether the proportion of tumours expressing SOX2 is increased.

The IVL+ basal population's bias towards asymmetric division and upward differentiation is part of its transcriptome-engrained commitment towards differentiation[1,4], likely contributing to its tumour-resistant phenotype. This resistance is also seen in basal cell carcinoma, where only the stem-cell population and not the progenitors have been proposed to initiate tumorigenesis[6]. However committed progenitors can trigger skin tumorigenesis once they re-acquire stemness features, induced by SURVIVIN expression in basal cell carcinoma[6] or by SOX2 in cSCC, as shown here.

The mechanism by which oncogenic signals activate SOX2 or related stemness programmes remains unclear, but MAPK or MYC-dependent pathways are likely candidates. Understanding not only the tissue-specific mechanisms that drive transformation but also the cell-specific vulnerabilities that govern progenitor state and maintenance in homoeostasis and tumorigenesis will be critical for developing therapeutic strategies aimed at halting tumour initiation at its cellular origin.

In summary, we introduce genetically driven in vivo models of cSCC to dissect tumour susceptibility in basal epidermal populations. Oncogene expression in the K14/K5+ stem cell population led to rapid tumour development, whereas IVL+ committed progenitors remained resistant to transformation. Nevertheless, SOX2 rendered this population permissive to tumorigenesis, preventing delamination and reducing apoptosis[32,33], by inducing a stem-like transcriptional state that mirrors the tumour-primed K14/K5+ population. Despite differences in cell of origin or oncogenic driver, we uncovered a shared transcriptional programme underpinning transformation, also extensible to other cSCC models. These findings highlight SOX2 as a critical switch that unlocks the tumorigenic potential of IVL+ differentiation-committed basal progenitors.

## Methods

### Genetically engineered mouse models and husbandry

Animal experiments conducted at the CRUK Scotland Institute Animal Facility were in accordance with the UK Home Office guidelines, under project licence PP3908577 and were reviewed and approved by the University of Glasgow Animal Welfare and Ethical Review Board (AWERB). Experiments conducted at the CRUK Manchester Institute were performed at Alderley Park Animal Research Unit, with breeding at the University of Manchester incubator breeding facility, in accordance with UK Home Office regulations, under project licences PE4369EDB and P671A5B06 and reviewed and approved by the CRUK Manchester Institute's AWERB. Mice were housed in accordance with UK Home Office Regulations, maintained in a specific pathogen-free facility under a 12-h light-dark cycle at a constant temperature between 19 and 23 °C and 55 ±10% humidity and given drinking water and fed standard chow diet *ad libitum*. Male and female mice were used throughout the study. Animals were monitored regularly until terminally euthanised by schedule 1, cervical dislocation, at clinical endpoint: either when the cumulative tumour burden had reached a

maximum volume of 1500 mm³ or when a single tumour exceeded 15 mm in diameter, weight loss >20%, or any other signs of ill health and distress. The aforementioned limits were not exceeded. For all mouse studies, no formal randomisation was performed and researchers were not blinded to the mouse genotypes. No exclusions were performed and all mice included in experimental cohorts were included in the analysis.

Genotyping was performed by Transnetyx according to the previously published protocols provided in the references for each allele. Mice were maintained on a C57BL/6J background, carrying the following alleles/transgenes: *Ivl*-CreERT2[15], *Krt5*-CreERT2[38], *Krt14*-CreERT2[39], BRAF[V600E40,41], HRAS[G12V42], *Rosa26LSL-tdRFP*[43], *Sox2*[fl44] and *Rosa26LSL-SOX2-IRES-eGFP*[45]. Archived tumour blocks [from[14]] included the additional alleles: *Lgr5*-CreERT2[46], *Kras*[G12D47], *Notch2*[fl48] and *Trp53*[fl49].

Genetic recombination was induced in mice of either sex 8–12 weeks of age by topical application to the shaved backs of the mice of 100 μL tamoxifen (10 mg/ml in ethanol; T5648, Sigma) in models driven by Ivl-CreERT2, or 4 μl 4-hydroxytamoxifen (20 mg/ml in ethanol; H6278, Sigma) in models driven by *Krt5*-CreERT2 and *Krt14*-CreERT, unless otherwise stated. This was repeated three more times over 8 days.

## UV irradiation

UV irradiation experiments were conducted at the CRUK Manchester Institute husbandry facility following the UK Home Office regulations under project licences PE4369EDB and P671A5B06. In cohorts exposed to UVR, mice were subjected to UVR exposure once per week for 4 weeks, beginning 4 weeks after the final tamoxifen application. For this process, mice were anaesthetised by intraperitoneal injection with 1 mg/kg Domitor and 100 mg/kg ketamine, with 5 mg/kg Antisedan anaesthetic reversal. The backs were shaved and a black cloth was used to cover regions that were to remain unexposed. The Waldmann UV181 unit with UV6 broad wavelength (280–380 nm) lamp was used for UVR exposure and the intensity was tested regularly with a USB2000+ spectroradiometer (Ocean Optics). The UV dose used was 0.6 kJ/m², which equated to 3 min and 14 s of exposure. Following irradiation, E45 moisturising cream was topically applied to the back.

## Tumour fragment implantation

Tumour implantation experiments were conducted at the CRUK Manchester Institute husbandry facility following the UK Home Office regulations under project licences PE4369EDB and P671A5B06. A tumour derived from a K5:BRAF[V600E] mouse was resected upon clinical endpoint under a laminar flow hood aseptically. The tumour was cut into 2–3 mm³ pieces for immediate implantation on anesthetised 6–8 week-old C57/6J littermates and NSGII2 mice. Briefly, recipient animals were anesthetised with isoflurane and kept on a heated stage. A tumour piece was implanted subcutaneously between the skin and the peritoneal wall. The surgical incision was closed using a surgical clip. Rymadyl/Carprofen analgesic was given at 4 mg/kg. Mice were monitored until consciousness was regained and then monitored daily. The surgical clip was removed 7 days post-surgery.

## In vivo drug treatments

In vivo drug treatments were conducted at the CRUK Scotland Institute Animal Facility following the UK Home Office guidelines, under project licence PP3908577. Skin inflammation was induced by topically applying 150 μl of TPA (31.25 μg/ml) in acetone (Sigma-Aldrich 16561-29-8) to shaved dorsal skin three times a week until tumour signs appeared. The BRAFi dabrafenib was administered by daily oral gavage (30 mg/kg in 100 μl) during the experiment.

## Immunohistochemistry and immunofluorescence

Organs collected in 10% neutral buffered formalin were stored at room temperature for 20–28 h, followed by transfer to 70% ethanol and storage at 4 °C until processing. All haematoxylin and eosin (H&E),

immunohistochemistry, co-immunofluorescence and in situ hybridisation staining were performed on 4-μm formalin-fixed paraffin-embedded sections (FFPE) which had previously been heated at 60 °C for 2 h.

The following antibodies were used on a Leica Bond Rx autostainer: KRT5 (905501, Biolegend), Ki67 (12202, Cell Signalling), SOX2 (14962, Cell Signalling), cleaved CASP3 (9661, Cell Signalling) and pSTAT3 (9131, Cell Signalling). All FFPE sections underwent onboard dewaxing (AR9222, Leica) and epitope retrieval using ER2 solution (AR9640, Leica) for 20 min at 95 °C. Sections were rinsed with Leica wash buffer (AR9590, Leica) before peroxidase block was performed using an Intense R kit (DS9263, Leica) for 5 min. Sections were rinsed with wash buffer before primary antibody application at an optimal dilution (KRT5, 1/1500; Ki67, 1/1000; pSTAT3, 1/100; SOX2, 1/200; CASP3, 1/500). The sections were rinsed with wash buffer before the application of anti-rabbit EnVision HRP-conjugated secondary antibody (K4003, Agilent) for 30 min. The sections were rinsed with wash buffer, visualised using DAB and counterstained with haematoxylin from the Intense R kit.

FFPE sections for KRT14 (ab7800, Abcam), c-MYC (ab32072, Abcam), RFP (600-401-379, Rockland), CD34 (119302, Biolegend), cPARP (ab32064, Abcam) and SOX9 (AB5535, Millipore) staining were stained on an Agilent autostainer Link48. FFPE sections were loaded into an Agilent pre-treatment module to be dewaxed and undergo heat-induced epitope retrieval (HIER) using a High pH target retrieval solution (K8004, Agilent). After HIER, the sections were rinsed in FLEX wash buffer (K8007, Agilent) before being loaded onto the Agilent autostainer. The sections underwent peroxidase blocking (S2023, Agilent) for 5 min and were rinsed with FLEX buffer. The primary antibody application was at an optimised dilution (KRT14, 1/300; c-MYC, 1/800; RFP, 1/1000; SOX9, 1/500; CD34, 1/100; cPARP 1/1000). Sections were washed with FLEX buffer before application of anti-rabbit EnVision secondary antibody for 30 min. Sections were rinsed with FLEX wash buffer before applying Liquid DAB (K3468, Agilent) for 10 min. Sections were washed in water and counterstained with haematoxylin 'Z' (RBA-4201-00A, CellPath).

In situ hybridisation detection for *Anxa1* (509298), *Ivl* (422538), *Ly6a* (427578), PPIB (313918; positive control) and dapB (312038; negative control) (all Bio-Techne) mRNA was performed using RNA-Scope 2.5 LSx (Brown) detection kit (322700; Bio-Techne) according to the manufacturer's instructions. H&E staining was performed on a Leica autostainer (ST5020). Sections were dewaxed in xylene, taken through graded ethanol solutions and stained with haematoxylin 'Z' (RBA-4201-00A, CellPath) for 13 min. Sections were washed in water, differentiated in 1% acid alcohol, washed and the nuclei stained blue in Scott's tap water substitute (in-house). After washing with tap water, sections were placed in Putt's eosin (in-house) for 3 min. To complete H&E, IHC & ISH staining, sections were rinsed in tap water, dehydrated through a series of graded alcohols and placed in xylene. The stained sections were coverslipped in xylene using DPX mountant (SEA-1300-00A, CellPath).

Sections for KRT14 and RFP co-IF staining were loaded onto a Leica Bond Rx autostainer. The FFPE sections underwent on-board dewaxing and epitope retrieval using ER2 solution for 20 min at 95 °C. Sections were rinsed with Leica wash buffer (AR9590, Leica) before application of 10% normal goat serum (X090710-8, Agilent) for 30 min. The sections were rinsed with Leica wash buffer before application of anti-RFP antibody at 1/1000 dilution for 1 h. Sections were rinsed with Leica wash buffer and goat anti-rabbit IgG 488 (A11034, Invitrogen) secondary antibody diluted 1/250 for 30 min. After rinsing with Leica wash buffer KRT14 antibody was applied at 1/750 dilution for 1 h. Sections were rinsed with Leica wash buffer and goat anti-mouse IgG 647 secondary antibody (A21236, Invitrogen diluted) 1/250 for 30 min before application of DAPI (MBD0015, Sigma-Aldrich). To complete the staining, sections were mounted

using ProLong Diamond antifade mountant (P36970, Thermo Fisher Scientific).

Immunohistochemistry images were acquired using a SCN400F slide scanner (Leica Microsystems) at ×20 or ×40 magnification. Confocal images were collected on a Zeiss 710 point-scanning confocal microscope, built on an inverted Zeiss Axio Imager. Z2 stand. Images were acquired using an EC Plan-Neofluar 40x/1.30 Oil and a confocal pinhole diameter of 107 μm. Multi-channel images were captured sequentially: DAPI (nuclear marker) using 405 nm excitation and 410–481 nm emission bandwidth, IgG 488 using 488 nm excitation and 514–582 nm emission and IgG 647 using 633 nm excitation and 638–747 nm emission. Images were collected with a 1× zoom with an image size of 2320 × 2320 pixels, yielding a pixel size of 92 × 92 nm and a 1.38 μs pixel dwell time. Z-stacks were collected using a step size of 2 μm. Images were acquired using the software Zen LSM 2.1 Black (Zeiss).

Quantification of RFP+ and CASP3+ cells for IHC-stained samples was performed using the HALO Image Analysis Platform version 3.6.4134 (Indica Labs, Inc.) to quantify the optical density of cellular staining after manual annotation of the epidermal layer and training to detect skin epithelial cells.

### RNAseq sequencing and analysis

RNA was extracted from fresh frozen tumour samples using an AllPrep DNA/RNA kit (74104, Qiagen) according to the manufacturer's instructions. Indexed poly(A) libraries were prepared using 200 ng of total RNA and 14 cycles of amplification with the Agilent SureSelect Strand-Specific RNA Library Preparation Kit for Illumina Sequencing (G9691B, Agilent). RNA polyA libraries were sequenced using NovaSeq 6000 XP SP, 200 cycles, paired-end reads (2 × 101 bp), 20–30 million reads. RNA preparation and sequencing of mouse samples were performed at the CRUK Manchester Institute histology core and sequencing facility.

RNA preparation and sequencing of Ivl:BRAF$^{V600E}$ and Ivl:BRAF$^{V600E}$-SOX2$^{LSL}$ tumours (Supplementary Fig. 10) were performed by GENEWIZ Azenta. rRNA depletion was performed before library preparation. Approximately thirty million paired-end reads (2 × 150 bp) per sample.

For all samples raw sequence quality control was performed using FastQC (v0.11.8) before and after removing adaptors and low-quality base calls (Phred score <20) using TrimGalore with default options (v0.6.6). Trimmed reads were aligned to GRCm38, release 100 using Hisat2 (v2.1.0). Gene counts were subsequently estimated using featurecounts (subread/1.6.3). Sva::ComBat-seq was used for batch correction (v3.52.0). For all samples, after removing transcripts without a minimum of 5 reads in at least one sample, the differential expression analysis between mouse tumours and skin was performed using the R package DESeq2 (v1.44.0). The resultant p-values were corrected for multiple comparisons using the Benjamini–Hochberg approach. The following additional R packages were used for downstream analysis: pheatmap (v1.0.8), fgsea (v1.30.0), GSVA (v4.5), enhancedVolcano (v1.22.0), clusterProfiler (v4.12.6), dplyr (v1.1.4), tidyr (v1.3.1), AnnotationHub (v3.12.0), ggvenn (0.1.10), AnnotationDbi (v1.66.0), mixOmics (v6.26.0).

Digital sorting and transcriptome deconvolution was performed using CIBERSORTx[50] using a signature matrix generated from the single-cell RNAseq dataset from Joost et al.[24] (GEO: GSE129218). CIBERSORTx was run online in relative mode to compare relative cellular fractions, using B-mode for batch correction and 100 permutations.

The 'cSCC MAPK signalling shared' signature derived from this work was generated by selecting the commonly upregulated and differentially expressed genes (log$_2$ fold change >3 and padj <0.01) obtained from Ivl:BRAF$^{V600E}$, K5:BRAF$^{V600E}$ and K14:BRAF$^{V600E}$ tumours

compared to normal skin from control litter mates (Supplementary Data 1). The 'SOX2_tumorigenesis' signature derived from this work was generated by selecting the top 150 differentially expressed genes (log$_2$ fold change >2 and padj <0.01) between Ivl:BRAF$^{V600E}$ and Ivl:BRAF$^{V600E}$-SOX2$^{LSL}$, of which 139 genes were converted to human orthologs (Supplementary Data 2).

### Statistics and reproducibility

Statistical analysis was performed using GraphPad Prism (v.10.0.1) for Windows. Normality and lognormality tests were used to establish the appropriate significance test, followed by a statistical test to compare the means. To compare the means of two groups, a two-tailed Student's $t$ test was performed. All error bars shown are mean ± standard error of the mean. Micrographs showing representative images refer to at least three independent mice unless stated otherwise. No statistical method was used to predetermine sample size. No data were excluded from the analyses, experiments were not randomised and the investigators were not blinded to allocation during experiments and outcome assessment. Mice allocation into experimental groups was random. Differences were considered statistically significant at $*p < 0.05$, $**p < 0.01$, $***p < 0.001$ and $****p < 0.0001$.

### Reporting summary

Further information on research design is available in the Nature Portfolio Reporting Summary linked to this article.

## Data availability

The data discussed in this publication have been deposited in NCBI's Gene Expression Omnibus. The accession numbers for the RNA-seq data reported in this paper are NCBI GEO GSE280236 and GSE303703. Source data are provided with this paper. The remaining data are available within the article, Supplementary Information or Source data file. Source data are provided with this paper.

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

## Acknowledgements

We thank the Core Services and Advanced Technologies at the Cancer Research UK Scotland and Manchester Institutes, particularly the Biological Services Unit, Histology Service and Molecular Technologies. We also thank Claire Mitchell and the BAIR Biological Advance Image Resource service at Cancer Research UK Scotland Institute. We thank Kevin Haigis (Dana-Farber Cancer Institute) for providing the HRAS^G12V mouse model. We are grateful to Sarah Ressel and Shan Quah for the critical review of the manuscript and to Nathalie Sphyris for expert writing advice (Cancer Research UK Scotland Institute). We would also like to thank Catherine Winchester for the research integrity review of the manuscript. This work was supported by Cancer Research UK Cancer Grand Challenges and The Mark Foundation for Cancer Research to the SPECIFICANCER team (A29055 & A27412) and by CRUK core funding to the Cancer Research UK Manchester Institute (C5759/A27412) and the Cancer Research UK Scotland Institute (C5759/A31287). P.P.C. and C.C. were funded by the Cancer Research UK Grand Challenge SPECIFICANCER Consortium. G.K., C.A.F., R.A.R., P.C., T.J. and O.J.S. were supported by CRUK Scotland Institute core funding (A17196, A31287).

## Author contributions

P.P.C., A.C., R.M. and O.J.S. designed experiments and interpreted results. P.P.C., C.C., G.K., C.A.F., R.A.R., T.J.S. and P.C. performed experiments and analysed results. P.P.C. analysed publicly available human cancer data sets. P.P.C. processed and analysed the RNA sequencing data. G.J.I. contributed to discussions and analysed results. P.P.C., A.C. and O.J.S. wrote the paper and reviewed and discussed the drafted manuscript. All authors contributed to the manuscript.

## Competing interests

The O.J.S. laboratory receives funding from Novartis, Cancer Research Technology (Cancer Research Horizons), Boehringer Ingelheim and AstraZeneca for other unrelated projects. R.M. is the founder and CEO of MyT Bioscience Ltd and the founder, Director and CSO of Oncodrug Ltd. The remaining authors declare no competing interests. Other authors declare no competing interests.
