## [Transparent Peer Review File · Nature Communications]

SOX2 confers tumour permissiveness in a specific skin progenitor population.

Corresponding Author: Professor Owen J Sansom

Version 0:

Reviewer comments:

Reviewer #1

(Remarks to the Author)

Melanoma patients treated with BRAF inhibitors commonly develop cSCCs due to paradoxical activation of the RAF signaling pathway. In this manuscript, the authors activated oncogenic BRAF or HRAS with K14-CreER or Inv-CreER to model the etiology of these tumors in mice. The authors identify a tumor-prone and tumor-resistant population and suggest that tumor-prone cells require SOX2 activation to develop into tumors. The models and data are overall well presented, but the manuscript lacks critical controls to relate findings to previously published results. It is also unclear whether and why Dabrafenib, resulting in paradoxical RAF activation, was only used in Fig. 1 or also in subsequent figures, and analyses of human tissues should have included and compared tumors initiated with and without Dabrafenib treatment. The manuscript will require more work as detailed below, before it can be published.

Major comments and/or concerns:

Oncogenic Ras has been initiated previously in K5/K14 (basal) and Inv (differentiating) skin epithelial cells, showing differences in tumor initiation similar to what has been described in this manuscript. Consistent with these reports, tumor-primed and tumor-resistant populations have been described, limiting the novelty of this work. However, this manuscript strengthens and extends prior work by comparing the transcriptomes of tumors developing in the tumor-primed and tumor-resistant lineages and identifying SOX2 as a transcription factor uniquely upregulated and required in the tumor-resistant population for tumorigenesis. This is consistent with a prior report identifying SOX2 in a few basal and more suprabasal cells in early-stage tumors (PMID: 25077433).

The authors should stain for IvI in the K14CreER and IVL-creER models to clarify if IVL is already expressed in some basal keratinocytes by staining the tissue with K14 and IVL antibodies and by performing IvI RNA-in situ hybridization. This is important to validate the faithfulness of the lineage tracing approach.

There is a disconnect between the Dabrafenib treatment results in Fig. 1 and subsequent figures. In Fig.1 it seems that Dabrafenib treatment is necessary to initiate tumor growth in Hras G12V/+ tumors. In subsequent figures, it seems Dabrafenib was not used, and now the authors use HrasG12V or BrafV600E alone. It is unclear how these subsequent figures relate to the paradoxical RAF activation and its role in promoting cSCCs in patients if Dabrafenib wasn't used. If Dabrafenib was used, it needs to be made clearer in the text and the figure legends. The paradoxical RAF activation is thought to stimulate dimerisation and activation of wild-type CRAF and it is more pronounced in cells expressing oncogenic Ras, which promotes RAF dimerization and activation of the MAPK pathway. The authors should also demonstrate MAPK activation within the activated K14 and Inv clones, ideally by also staining for CD34 and SOX2, as this would clarify the sequence of events and position the work better in context with prior studies and their models where Hras G12V was homozygous or heterozygous with and without Dabrafenib treatment.

The statement that SOX2 overexpression is sufficient to drive tumorigenesis from the tumor-resistant population is misleading or incorrect, as an oncogenic mutation is required to initiate tumors. Later in the text, the authors state they were testing whether SOX2 is needed. They further state that SOX2 overexpression accelerated tumor onset, but SOX2 overexpression alone showed no phenotype. These contradictions should be corrected in the text.

It is unclear where the tumors originate in the K5 and K14 models, as these markers are expressed in the epidermis and the hair follicle.

Inv-cre, in contrast, is only expressed in the interfollicular epidermis. This paper suggests that K14-cre-initiated tumors don't express SOX2. This is surprising because a prior study identified SOX2 in K14-positive cells expressing CD34 (PMID: 24909994). It is unclear whether and how the authors of this manuscript excluded cells within the hair follicle lineage that express K5/K14 and CD34. The authors should stain their tissues for CD34 and determine whether CD34 positive epidermal or tumor cells were included or excluded from their study.

The authors suggest that SOX2 is not required in tumors initiated with K14Cre-BrafV600E, and they don't express SOX2 in contrast to InvCre-initiated tumors. Previous studies by Boumahdi et al. (PMID: 24909994) showed that conditional ablation of SOX2 with K14cre restricted DMBA/TPA-initiated tumor growth. It is unclear how the authors explain this discrepancy.

The authors claim that SOX2 overexpression in the skin controls stemness. However, they do not specify what they mean by this term. They seem to see more transcriptional changes when they express SOX2 in adult compared to juvenile human keratinocytes. Whether these keratinocytes were derived from the same body site, and body site-specific differences in gene expression programs, need to be considered. Another possibility is the difference in SOX2 overexpression, which is 7 or 12-fold on a log2 scale in juvenile and adult keratinocytes. It seems logical that the weaker SOX2 overexpression in juvenile keratinocytes would cause fewer gene expression changes.

The authors overexpress SOX2 in adult and juvenile keratinocytes and present data suggesting that SOX2 restores a juvenile expression pattern. A previously published study (PMID: 30045979) performed a similar experiment, implying that cutaneous keratinocytes acquire an oral keratinocyte expression program prone to improved wound healing responses. Do the authors also see an oral program, and if so, how does the oral program compare to the juvenile program? It is unclear why the authors compare the expression data from human keratinocytes to expression profiles from mice. The authors should clarify their rationale for this more clearly.

The authors analyze a tissue microarray of 250 patients and find ~25% of tumors expressing mid-high SOX2 levels, as previously reported and referenced by the authors. It seems crucial for the narrative of this paper to test whether this distribution is affected by Dabrafenib treatment. As presented, it mostly confirms SOX2 expression data shown by others, which is nice but less impactful than providing a clearer association with paradoxical RAF activation, if that exists.

MINOR weaknesses and suggestions:

The introduction states HRAS mutations are uncommon in cSCC developing spontaneously in patients, but common in melanoma patients treated with BRAF inhibitors. While 9-16 % frequency is listed for spontaneous, it is missing for BRAF inhibitor-treated patients. The authors should add this information.

SOX2 expression in DMBA/TPA-induced tumors initiating in interfollicular epidermis was previously reported. It was also reported that tumors initiating in LGR5+ cells don't express SOX2 or EpCAM and are more mesenchymal. The authors should reference these reports when describing their findings in lines 266-268.

Was the Western blot in Fig. 7A probed simultaneously with SOX2 and ACTB antibodies? The blots look unconventional. Protein markers are missing and should be added.

Reviewer #2

(Remarks to the Author)

In this manuscript, Centeno and colleagues aim at uncovering the effect of BRAF activation in skin squamous cell carcinoma formation, upon activation of this oncogene in two different cell populations located in the skin basal layer (keratin 14 (K14)-expressing and Involucrin (Ivl)-expressing cell populations). It has been previously reported that those two cell populations differ in the competence to initiate basal cell carcinoma (Sánchez-Danés et al, Nature 2012). In that study and in a more recent study (Canato et al, Cancer Discovery 2025), it was shown that one genetic alteration (Hedgehog activation) is required to initiate tumour from the K14-expressing population but two genetic alterations are needed in the Ivl-expressing cells to initiate a tumour (TP53 deletion or Survivin overexpression). In addition, the key role of Sox2 in papilloma and cutaneous skin carcinoma was shown using genetic and chemical-induced mouse models of squamous cell carcinoma (Boumahdi et al Nature 2014) . However, the synergy between Sox2 and BRAF activation in squamous cell carcinoma formation has not been yet explored.

In this manuscript, the authors first study whether BRAF inhibitor treatment synergizes with HRAS G12V activation in K14 and Ivl-expressing cells to initiate cSCC. They found that BRAF inhibitor treatment leads to tumour formation in both models, with faster tumour onset in the K14-CreER model compared to the Ivl-CreER model, while HRAS G12V alone was not sufficient to initiate skin tumour formation. Next, the authors tested whether BRAF V600E activation in K14, K5 and Ivl-expressing cells was sufficient to initiate cSCC, and found that BRAF activation in K14 and K5 -expressing cells lead to the formation of rare tumours with a median tumour onset of 9-19 days, while Ivl-expressing cells lead to tumours with a median onset of 125 days. In order to understand if the tumours derived from K14, K5 and Ivl-expressing cells were similar/different in terms of their transcriptomic profile, the authors performed bulk RNAsequencing in unsorted tumours and found that the tumours arising from K14 and K5-expressing populations were really similar at the transcriptome level, and those tumours

had an overlap of around 80% with the genes upregulated in the *lvl*:BRAF V600E tumours. Sox 2 was one of the genes highly expressed in the *lvl*:BRAF V600E tumours compared to K15&K5:BRAF V600E tumours. In order to assess the role of Sox2 in cSCC tumorigenesis upon BRAF V600E activation, the authors used genetic mouse models to delete Sox2 and activate BRAF V600E in the K14 and *lvl*-expressing population. The authors found that Sox2 deletion in K14 model did not have any impact in tumour latency, but it did delay tumour formation in the *lvl* model. Finally, they show that Sox2 overexpression together with BRAF V600E activation in the *lvl*-expressing cells lead to faster tumour onset compared with BRAF V600E activation alone.

I consider that the topic of this study is of interest, but the experiments performed and data provided are not sufficient to substantiate the authors' claims. In addition, it is difficult to follow the rationale of the study, specially it is not well explained why the authors decided to end the study describing the expression of Sox2 in young vs juvenile keratinocytes. This last part is totally unrelated to the rest of the study and could probably constitute a study on its own.

There are some key points that need to be improved and/or clarified:

1-The authors do not describe whether the tumours originated from the different genetic mouse models are papillomas and/or carcinomas. This should be added for each of the genetic models used, as it would be helpful to understand whether the Kaplan-Meier curves presented correspond to papillomas or carcinomas. In general the H&N images and immunohistochemistry stainings in all figures are poor quality and it is difficult to assess if the lesions are papillomas and carcinomas, high resolution images and magnifications are required.

2. In Figure 1 the authors show that HRAS activation synergizes with BRAF inhibitor treatment to induce cSCC formation. In the manuscript the authors claim that this is due to paradoxical MAPK signalling activation (line 87, line 813), the authors should support this claim providing experimental proof of this claim or alternatively, tune down the statement.

3. In Figure 2 panel E the authors count the number of tumorigenic lesions formed in the different mouse models used. However, for each mouse model a different time point was used, this should be stated in the manuscript text. Otherwise, the authors should study the number of tumours formed in K14:BRAF or K5:BRAF models by resecting the tumours formed until they reach the same timepoint as the *lvl*-models. Potentially, the K14 or K5 models might lead to the same number or more tumours formed than the *lvl* in a given period of time (the same for all genotypes).

4. It is difficult to interpret Figure 3 in light of the results of Figure 2. In Figure 2 it is shown that in the K14:BRAF V600E model tumours arise with a median tumour onset of 10 days, while in Figure 2 D-F there is no tumour being shown but only wild type skin at Day 8 and Day 15 (missing Day 28 image and quantifications). It would be helpful to understand the rationale of the experiments performed in Figure 2 and why they decided to focus on wild-type skin and not tumorigenic lesions from the K14 model.

In addition, it would be interesting to understand what is the mechanism that is leading to the disappearance of the hyperplastic lesions in the *lvl*:BRAF V600E-tdRFP model.

5. In Figure 3 if the authors compare the effect of BRAF V600E activation in the K14 and *lvl* population using lineage tracing and clonal dynamics analysis. The authors use a non-clonal/saturation dose for the K14 model and a clonal dose for the *lvl* model. However, in order to be able to directly compare the two models, the authors should be targeting in both models the same number of cells as previously performed (Mascre et al Nature 2012, Sánchez-Dané et al Nature 2016).

6. In Figure 4 the authors perform a bulk RNA sequencing of BRAF V600E tumours from the *lvl*, K14 and K5 models. Are those tumours papillomas or carcinomas and are there differences in the tumour stage between the *lvl* and K14/K5 analysed?

The authors perform bulk RNA sequencing in tumour samples, that are not composed of pure tumour cells, but contain both tumour cells and tumour microenvironment. Are the lack of major differences observed between K14&K5 vs *lvl* tumours due to an enrichment of tumour microenvironment cell populations that mask the differences in the transcriptional profile of the *lvl* vs K14&K5 tumour cells? A possible way to uncover this relevant point would be to FACS sort tumour cells using the BRAF V600E model: RFP (used in Figure 3) and perform bulkRNAseq in those samples.

7. In Figure 5 it is shown that SOX2 expression in the *lvl*-expressing cells synergizes with BRAF V600E and accelerates tumour formation, but not that "SOX2 overexpression is sufficient to drive tumorigenesis" (line 280). This statement should be corrected, as it is not an accurate description of the data presented.

In order to understand if the effect of Sox2 expression is specific to the *lvl*-expressing cells, the authors should also show what happens upon Sox2 overexpression in the K14:BRAF V600E model.

It would be really interesting to understand if Sox2 overexpression promotes the progression of papillomas to cSCC in the BRAF model.

8. The use of "*lvl*-expressing cells" throughout the manuscript instead of "tumour-resistant population" is more accurate and might not lead to confusion.

9. The title does not describe the current data and should be revised.

Other specific comments:

1- In Figure 1 the scheme in A is misleading, it seems that the animals are both K14CreER and *lvl*-CreER. Same comment

for Figure 2A and Figure 3A. Please use the same strategy depicted in Figure 6 A.

2. Abstract, 3rd sentence. "How the skin tolerates these mutations in different populations... is poorly understood" this statement is not correct as numerous studies from Blanpain lab and Lowry lab among others have contributed to the understanding of the effect of different mutations in cSCC formation.
3. Introduction. Line 66, it should be described that the IVL and K14 cell populations show different tumour initiating abilities (Sánchez-Danés et al Nature 2016), as this point is really relevant for the present study.
4. Figure 4 A, is there a reason for the big dispersion observed within the samples of the control group?
5. Figure 4C, K14_K14 in the title should be corrected for K5_K14

I recommend addressing the points mentioned above before the paper can be considered for publication in Nature Communications.

Version 1:

Reviewer comments:

Reviewer #1

(Remarks to the Author)

Centeno et al. addressed several issues raised by reviewers in their initial submission and omitted some of their results. However, several new or remaining issues need to be addressed before the revised manuscript can be accepted.

1. The authors state in the second sentence of their abstract that "rapid cSCC development occurs in melanoma patients treated with BRAF inhibitors. To model this in mice, we induced MAPK hyperactivation in two epidermal populations: stem-like (K5/K14) and differentiation-committed (IVL+) cells." It is unclear how these mice would model BRAF inhibitor treatment, whether the mice still require treatment with BRAF inhibitors, and other related details.
2. The abstract also refers to SOX2 as a pioneer factor. Although this may not be incorrect, this paper does not address a possible pioneer factor function of SOX2. It appears again in line 416. Although a function for SOX2 as a pioneer factor in SCCs has not been demonstrated, Sastre Perona et al, Cell Stem Cell 2019, showed that SOX2 and PITX1 bind to chromatin that is accessible in cSCC but not epidermal progenitor cells. The authors should include this in their discussion and cite this work when arguing a potential role of SOX2 as a pioneer factor that could rewire the epigenome.
3. Line 71-72: .. revealing that tumors retain the transcriptional memory of their cell of origin. How so? It is unclear what the authors refer to with this sentence, and it is unclear from the results which mechanisms are retained and which are not. The authors should be clearer.
4. lines 137-143: Why does their IVL-BRAF model develop tumors at multiple sites on the body even without TAM treatment, whereas the K14 model develops tumors only at the spot that was treated? Is this due to leaky or unspecific expression? The authors should explain.
5. The authors refer to their tumors throughout as cSCC, but it is unclear whether some of their tumors are benign papillomas.
6. The authors show that SOX2 appears first in differentiation-committed IVL+ cells. Sastre Perona et al, Cell Stem Cell 2019 also reported SOX2 expression to first appear in some suprabasal papilloma cells, before it appeared in most basal cSCC cells. The authors should reference this work.
7. lines 240-244: The bulk RNA-seq data must be viewed with caution, as the authors have analyzed total tumor tissue rather than tumor epithelial cells in this study.
8. lines 270-275: could identify markers of the cell of origin of cSCC tumours..... SOX2 emerged as a key transcriptional difference, being highly enriched in IVL This paragraph is misleading because SOX2 is not expressed in the cell of origin or normal skin epithelial cells, as reported by several studies. The authors may want to refer to the IVL lineage rather than the cell of origin to avoid confusion.
9. Why is SOX2 referred to as a super pioneer factor and not a pioneer factor? This terminology seems odd.
10. line. 286: "We did not detect SOX2 expression or CD34 in either of the models at earlier time points." This sentence seems out of place in this paragraph. However, it supports my point (8) that the authors shouldn't refer to the cell of origin if SOX2 is only detected at the experimental endpoint but not at earlier time points.
11. Lines 291-299 are convoluted and distractive. It starts by wanting to highlight the cell-specific role of SOX2 and then discusses SOX9, and ends up with a statement on differences in mRNA and protein of KRT4 and KRT13.
12. Line 356: "CD34 expression, marking a population of tumor-initiating cells, was observed from day 15....." " This paper doesn't test whether CD34-positive cells are tumor-initiating, and previous studies have demonstrated that CD34 expression is dynamic, and CD34+/CD49f+ and CD34-/CD49f+ cells form tumors at the same frequency, and their daughter tumors contain CD34+ and CD34- SCC cells. The interconversion between different cell states in SCCs has also been demonstrated with other cell surface markers and functional features. Therefore, it's unclear what point the authors want to make.
13. Line 366: .. the apoptosis hallmark described here was reported by Boumahdi et al Nature 2014. The authors should add a reference to their finding as it confirms Boumahdi's findings.
14. line 371-372: ".... IVL+ derived tumors are indistinguishable from those originating from a K14/5+ stem like population." - This statement must be wrong as the authors argue throughout that IVL+ express and depend on SOX2, whereas K14/5

derived tumors do not.

15. line 400: The sentence ends with shown here, but it is unclear what core set of skin transformation pathways the authors are referring to and where they are shown in this paper.

16. line 432 is an overstatement: This manuscript shows SOX2 promotes/is required for cSCC development in an IVL-BRAF model consistent with previous reports in lung cancer (Xu et al, Genes and Development 2014), but it does not show that the cells are being reprogrammed. To support this claim, the authors would need to profile the epigenomes of the respective cells of origin and their tumors to demonstrate that the cells were reprogrammed.

17. line 442: "... preventing delamination and reducing apoptosis" The authors should include references to Siegle et al Nat. Comm 2014 shows that SOX2 affects symmetric versus asymmetric renewal, which is presumably similar to delamination here, and Boumahdi should be referenced regarding reduced apoptosis.

Reviewer #2

(Remarks to the Author)

The authors have thoroughly addressed all my points by providing new experimental data and making substantial improvements to the manuscript.

Version 2:

Reviewer comments:

Reviewer #1

(Remarks to the Author)

The authors addressed my remaining concerns.

REVIEWER COMMENTS

Reviewer #1

Expertise in cutaneous squamous cell carcinoma, stem-like cells, SOX2, multi-omics and mouse models (Remarks to the Author):

Melanoma patients treated with BRAF inhibitors commonly develop cSCCs due to paradoxical activation of the RAF signaling pathway. In this manuscript, the authors activated oncogenic BRAF or HRAS with K14-CreER or Inv-CreER to model the etiology of these tumors in mice. The authors identify a tumor-prone and tumor-resistant population and suggest that tumor-prone cells require SOX2 activation to develop into tumors. The models and data are overall well presented, but the manuscript lacks critical controls to relate findings to previously published results. It is also unclear whether and why Dabrafenib, resulting in paradoxical RAF activation, was only used in Fig. 1 or also in subsequent figures, and analyses of human tissues should have included and compared tumors initiated with and without Dabrafenib treatment. The manuscript will require more work as detailed below, before it can be published.

Oncogenic Ras has been initiated previously in K5/K14 (basal) and Inv (differentiating) skin epithelial cells, showing differences in tumour initiation similar to what has been described in this manuscript. Consistent with these reports, tumor-primed and tumor-resistant populations have been described, limiting the novelty of this work. However, this manuscript strengthens and extends prior work by comparing the transcriptomes of tumors developing in the tumor-primed and tumor-resistant lineages and identifying SOX2 as a transcription factor uniquely upregulated and required in the tumor-resistant population for tumorigenesis. This is consistent with a prior report identifying SOX2 in a few basal and more suprabasal cells in early-stage tumors (PMID: 25077433).

We thank the reviewer for their constructive feedback and suggestions, which have significantly improved the manuscript. Please find our additional data and comments detailed below.

Major comments and/or concerns:

1. The authors should stain for IVL in the K14CreER and IVL-creER models to clarify if IVL is already expressed in some basal keratinocytes by staining the tissue with K14 and IVL antibodies and by performing IVL RNA-in situ hybridization. This is important to validate the faithfulness of the lineage tracing approach.

We thank the reviewer for the suggestions. Following their advice and to validate the faithfulness of the lineage tracing approach, we performed co-immunofluorescence using combinations of K14 and RFP, or IVL and RFP as suggested. IVL and RFP colocalised perfectly in our model, while K14 is expressed in all cells in the basal layer as expected. See Figure 3 and Supplementary Fig. 4. Pasted here for reference:

We also stained for IVL protein in early time points as suggested (Supplementary Fig.4), revealing similar results to RFP staining.

We also tried to use RNA-ISH; nevertheless, the RNA level was very low, so the figure was non-informative. Attached here for reviewers' reference.

2. There is a disconnect between the Dabrafenib treatment results in Fig. 1 and subsequent figures. In Fig.1 it seems that Dabrafenib treatment is necessary to initiate tumor growth in Hras G12V/+ tumors. In subsequent figures, it seems Dabrafenib was not used, and now the authors use HrasG12V or BrafV600E alone. It is unclear how these subsequent figures relate to the paradoxical RAF activation and its role in promoting cSCCs in patients if Dabrafenib wasn't used. If Dabrafenib was used, it needs to be made clearer in the text and the figure legends.

We apologise for the lack of clarity. Indeed, Dabrafenib was needed to promote tumours from HRASG12V het models. However, the BRAFV600E model did not require further promotion, and the tumours appeared spontaneously. We hypothesise that BRAF V600E is a much stronger activator of the MAPK signalling pathway (see comment below).

We have also clarified the text, figures, and figure legends. We have also added schemes as appropriate. See figures 1G and 2A for examples.

3. The paradoxical RAF activation is thought to stimulate dimerisation and activation of wild-type CRAF and it is more pronounced in cells expressing oncogenic Ras, which promotes RAF dimerization and activation of the MAPK pathway. The authors should

also demonstrate MAPK activation within the activated K14 and Inv clones, ideally by also staining for CD34 and SOX2, as this would clarify the sequence of events and position the work better in context with prior studies and their models where Hras G12V was homozygous or heterozygous with and without Dabrafenib treatment.

In order to confirm MAPK signalling activation, we have stained the skin with the downstream markers DUSP6 and pERK in HRAS, het, hom, TPA and in the presence of BRAFi (see Figure 1d, supplementary Figures 1B, 2B and 2E). This way, we have identified that a threshold level of MAPK signalling activation is required for transformation, only achieved with heterozygous HRAS^{G12V} and BRAFi combination, homozygous mutations of HRAS^{G12V}, or BRAF^{V600E}.

We have also stained our endpoint tumors and time point analysis for SOX2 and CD34 as suggested. Both being negative (see Supplementary Fig. 7). Nevertheless, we have seen CD34 activation in the SOX2 overexpressing model Figure 6i, which appears to synergise with SOX2 marking the population of “tumour initiating cells” previously reported by Blanpain’s Lab. We would like to thank you for the great suggestion that has helped us to clarify the mechanism.

4. The statement that SOX2 overexpression is sufficient to drive tumorigenesis from the tumor-resistant population is misleading or incorrect, as an oncogenic mutation is required to initiate tumors. Later in the text, the authors state they were testing whether SOX2 is needed. They further state that SOX2 overexpression accelerated tumor onset, but SOX2 overexpression alone showed no phenotype. These contradictions should be corrected in the text.

We apologise for the misleading text. This has been amended in the manuscript. Example corrections include: “The IVL+ tumour-resistant basal population depends on SOX2 for transformation”, “SOX2 accelerates transformation in the IVL+ tumour-resistant population”, “SOX2 overexpression in combination with MAPK activation is sufficient to initiate cSCC from the IVL+ tumour-resistant population” and “Overexpression of SOX2 renders the IVL+ tumour-resistant population susceptible to tumorigenesis”.

5. It is unclear where the tumors originate in the K5 and K14 models, as these markers are expressed in the epidermis and the hair follicle. *Inv-cre*, in contrast, is only expressed in the interfollicular epidermis. This paper suggests that K14-*cre*-initiated tumors don't express SOX2. This is surprising because a prior study identified SOX2 in K14-positive cells expressing CD34 (PMID: 24909994). It is unclear whether and how the authors of this manuscript excluded cells within the hair follicle lineage that express K5/K14 and CD34. The authors should stain their tissues for CD34 and determine whether CD34 positive epidermal or tumor cells were included or excluded from their study.

We thank the reviewer for highlighting this. In the present study, we did not exclude any cell population or tumour. MAPK-driven tumour formation in the K5/K14 population originated from the interfollicular epidermis, with hair follicles remaining normal throughout the study. Careful analysis of the histological H&E slides reveals epidermal thickening before tumour formation, accompanied by normal hair follicles, which remain normal once the tumour emerges in all our samples (Supplementary Fig. 1, Figure 2, and Figure 3). Additionally, from our previous experience, we know that expressing the BRAF^{V600E} oncogene in the hair follicle population (*Lgr5+*) is not enough to drive tumourigenesis, and further mutations in the TGF β receptor are required for transformation (Cammareri et al, 2016 & Bailey et al 2023). Although the fact that MAPK-driven tumours emerge only from the epidermis and not hair follicles is an interesting finding, it is beyond the scope of this paper.

Furthermore, we have conducted digital sorting of our bulk RNA-seq data and concluded that all our tumours are mainly composed of permanent epithelial cells (~70%) (Supplementary Fig. 5d). In comparison, cells related to the hair follicle are found in 10% or less of the total bulk tumour population. Moreover, no differences in tumour composition were found between tumours emerging from K5/K14 or *lvl*.

We are familiar with the study mentioned (Boumahdi et al, 2012), where the authors express Sox2 in the K14 population; nevertheless, they drove tumour formation using the DMBA/TPA protocol that triggers transformation from the *Lgr6* population as the cell of origin (Taylor et al, 2024, PMID 38815020). Thus, our results are in line with the cited study (Boumahdi et al, 2012), as SOX2 is indeed expressed in DMBA/TPA-derived tumours (Supplementary Figure 7) [see also comment below].

CD34 is normally expressed in fibroblasts and endothelial cells, although it has been proposed as a marker for cancer stem cells (Boumahdi et al, 2012). In our time-point analysis, we did not observe abnormal CD34 expression beyond the dermis (fibroblast-rich) (Supplementary Figure 7). It has also been proposed that SOX2 and CD34 will mark together a population of tumour-initiating cells, which is why we stained our Sox2 overexpressing tumours (*lvl*:BRAF:Sox2) for CD34 and found high protein expression (Figure 6) in contrast to K14 or *lvl* driven models that were negative for CD34 (Supplementary Figure 7). We thank the reviewer for this suggestion, as it has helped us to clarify the mechanism.

6. The authors suggest that SOX2 is not required in tumors initiated with K14Cre-BrafV600E, and they don't express SOX2 in contrast to *InvCre*-initiated tumors. Previous studies by Boumahdi et al. (PMID: 24909994) showed that conditional ablation of SOX2 with K14cre restricted DMBA/TPA-initiated tumor growth. It is unclear how the authors explain this discrepancy.

This is a very interesting point. Indeed, in our experiments, ablation of SOX2 in the K14 population didn't impact tumour onset or growth in the K14 population, while it delayed both in the Iv1 population (Figure 6).

As we have shown here, DMBA/TPA tumours show SOX2 expression (Supplementary Figure 7). The cited study, which drives tumours from the hair follicle Lgr6 population using the DMBA/TPA protocol, shows delayed tumour growth after SOX2 ablation in the K14 population. However, the Lgr6+ population also expresses K14+. Thus, this data suggests that the Lgr6+K14+ population will need SOX2 activation for transformation, similar to the Iv1 population (which also expresses K14). Meanwhile, the basal K14+ (IVL-/ LGR6-) population would not rely on Sox2 for transformation. This has been added in the discussion line 410. Together, we hypothesise that when tumours are being formed outside of the K14/5 basal stem cell-like population, either within the Lgr6+ population or within the IVL+ population, SOX2 is required for tumourigenesis to occur efficiently.

7. The authors claim that SOX2 overexpression in the skin controls stemness. However, they do not specify what they mean by this term. They seem to see more transcriptional changes when they express SOX2 in adult compared to juvenile human keratinocytes. Whether these keratinocytes were derived from the same body site, and body site-specific differences in gene expression programs, need to be considered. Another possibility is the difference in SOX2 overexpression, which is 7 or 12-fold on a log2 scale in juvenile and adult keratinocytes. It seems logical that the weaker SOX2 overexpression in juvenile keratinocytes would cause fewer gene expression changes.

We agree with the reviewer that this *in vitro* experiment was not conclusive, due to the different donor profiles (age, sex, body site, and Sox2 levels). Therefore, we have decided to remove it from the manuscript. Nevertheless, to further study the role of SOX2 in the Iv1 population, we have conducted timepoint experiments in Iv1:BRAF^{V600E}:SOX2^{LSL}, studying tumour formation, and different immunostainings for selected markers (Figure 6 and Supplementary Fig 9). We can see that SOX2 drives a stemness profile which includes lack of differentiation and increased basal layer retention of the IVL+ population, reduced apoptosis and tumour-initiating capabilities.

Moreover, we have conducted transcriptomic analysis comparing our *in vivo* Ivl-derived tumours with those overexpressing Sox2 (Supplementary Figure 10). Sox2 overexpressing tumours are enriched with epithelial-mesenchymal, STAT3, and immune activation pathways. Text related to this section has been substantially modified (see from line 312).

- The authors overexpress SOX2 in adult and juvenile keratinocytes and present data suggesting that SOX2 restores a juvenile expression pattern. A previously published study (PMID: 30045979) performed a similar experiment, implying that cutaneous keratinocytes acquire an oral keratinocyte expression program prone to improved wound healing responses. Do the authors also see an oral program, and if so, how does the oral program compare to the juvenile program? It is unclear why the authors compare the expression data from human keratinocytes to expression profiles from mice. The authors should clarify their rationale for this more clearly.

We agree with the reviewer that this *in vitro* experiment was not conclusive, due to the different donor profiles (age, sex, body site, and Sox2 levels), and it did not test our hypothesis. This data has now been substituted (see comment above).

- The authors analyze a tissue microarray of 250 patients and find ~25% of tumors expressing mid-high SOX2 levels, as previously reported and referenced by the authors. It seems crucial for the narrative of this paper to test whether this distribution is affected by Dabrafenib treatment. As presented, it mostly confirms SOX2 expression data shown by others, which is nice but less impactful than providing a clearer association with paradoxical RAF activation, if that exists.

Unfortunately, although there is genetic data available for the tumours that have arisen under BRAF inhibitor treatment (and these have high levels of HRAS mutations), there is no RNA data available. We had added this important caveat to our discussion (line 422). Given the finding that DMBA/TPA tumours also express SOX2, the finding that 25% of human SCCs express SOX2, and the different requirement for SOX2 in across models, as shown in our and other studies, it is likely that the role we described is conserved in human tumourigenesis. Moreover, we see the same association with anti-apoptotic pathways in the SOX2 overexpressing human tumourigenesis, suggesting that those tumours arising from a more differentiated population may need to overcome the apoptotic pathway.

MINOR weaknesses and suggestions:

The introduction states HRAS mutations are uncommon in cSCC developing spontaneously in patients, but common in melanoma patients treated with BRAF inhibitors. While 9-16 % frequency is listed for spontaneous, it is missing for BRAF inhibitor-treated patients. The authors should add this information.

Thank you for the suggestion. We have now added the data as follows: “However, HRAS-driven cSCC is commonly found in ~60% of melanoma patients arising within weeks of treatment with BRAFV600E inhibitors (BRAFi)”. Line 58.

SOX2 expression in DMBA/TPA-induced tumors initiating in interfollicular epidermis was previously reported. It was also reported that tumors initiating in LGR5+ cells don't express SOX2 or EpCAM and are more mesenchymal. The authors should reference these reports when describing their findings in lines 266-268.

We apologise for overlooking this. Reference has been added, now line 290.

Was the Western blot in Fig. 7A probed simultaneously with SOX2 and ACTB antibodies? The blots look unconventional. Protein markers are missing and should be added.

Thank you for noting this. Indeed, the blots were performed using the LICOR system, which enables the simultaneous detection of two proteins using fluorophores that emit light in different channels. Nevertheless, the new version of the manuscript does not contain Western blots.

Reviewer #2

Expertise in cutaneous squamous cell carcinoma development and cellular origin (Remarks to the Author):

In this manuscript, Centeno and colleagues aim at uncovering the effect of BRAF activation in skin squamous cell carcinoma formation, upon activation of this oncogene in two different cell populations located in the skin basal layer (keratin 14 (K14)-expressing and Involucrin (Ivl)-expressing cell populations). It has been previously reported that those two cell populations differ in the competence to initiate basal cell carcinoma (Sánchez-Danés et al, Nature 2012). In that study and in a more recent study (Canato et al, Cancer Discovery 2025), it was shown that one genetic alteration (Hedgehog activation) is required to initiate tumour from the K14-expressing population but two genetic alterations are needed in the Ivl-expressing cells to initiate a tumour (TP53 deletion or Survivin overexpression). In addition, the key role of Sox2 in papilloma and cutaneous skin carcinoma was shown using genetic and chemical-induced mouse models of squamous cell carcinoma (Boumahdi et al Nature 2014) . However, the synergy between Sox2 and BRAF activation in squamous cell carcinoma formation has not been yet explored.

In this manuscript, the authors first study whether BRAF inhibitor treatment synergizes with HRAS G12V activation in K14 and Ivl-expressing cells to initiate cSCC. They found that BRAF inhibitor treatment leads to tumour formation in both models, with faster tumour onset in the K14-CreER model compared to the Ivl-CreER model, while HRAS G12V alone was not sufficient to initiate skin tumour formation. Next, the authors tested whether BRAF V600E activation in K14, K5 and Ivl-expressing cells was sufficient to initiate cSCC, and found that BRAF activation in K14 and K5 -expressing cells lead to the formation of rare tumours with a median tumour onset of 9-19 days, while Ivl-expressing cells lead to tumours with a median

onset of 125 days. In order to understand if the tumours derived from K14, K5 and lvl-expressing cells were similar/different in terms of their transcriptomic profile, the authors performed bulk RNAsequencing in unsorted tumours and found that the tumours arising from K14 and K5-expressing populations were really similar at the transcriptome level, and those tumours had an overlap of around 80% with the genes upregulated in the lvl;BRAF V600E tumours. Sox 2 was one of the genes highly expressed in the lvl;BRAF V600E tumours compared to K15&K5;BRAF V600E tumors. In order to assess the role of Sox2 in cSCC tumorigenesis upon BRAF V600E activation, the authors used genetic mouse models to delete Sox2 and activate BRAF V600E in the K14 and lvl-expressing population. The authors found that Sox2 deletion in K14 model did not have any impact in tumour latency, but it did delay tumour formation in the lvl model. Finally, they show that Sox2 overexpression together with BRAF V600E activation in the lvl-expressing cells lead to faster tumour onset compared with BRAF V600E activation alone.

I consider that the topic of this study is of interest, but the experiments performed and data provided are not sufficient to substantiate the authors' claims. In addition, it is difficult to follow the rationale of the study, specially it is not well explained why the authors decided to end the study describing the expression of Sox2 in young vs juvenile keratinocytes. This last part is totally unrelated to the rest of the study and could probably constitute a study on its own.

We thank the reviewer for their constructive feedback; we have now produced additional data that both strengthens and clarifies the hypothesis. Their comments have substantially improved the manuscript. Please find our additional data and comments detailed below.

Major points

There are some key points that need to be improved and/or clarified:

1-The authors do not describe whether the tumours originated from the different genetic mouse models are papillomas and/or carcinomas. This should be added for each of the genetic models used, as it would be helpful to understand whether the Kaplan-Meier curves presented correspond to papillomas or carcinomas. In general the H&N images and immunohistochemistry stainings in all figures are poor quality and it is difficult to assess if the lesions are papillomas and carcinomas, high resolution images and magnifications are required.

Apologies for the lack of clarity. For the models to be comparable, the Kaplan-Meier (KM) curves correspond to tumour onset (tumour-free). Mice were routinely checked at least twice a week. Once the first macroscopical change in the skin was observed, the date was recorded and added to the Kaplan-Meier curve. All our KM curves represent the tumour-free survival from time of induction until time of the first lesion or macroscopic change in the skin. This has been clarified in line 103 as "Note that the tumour-free survival rate refers to the time from the first spotted lesion or macroscopic change in the skin" and also in Figure legend 1.

We have carefully analysed all the H&E histology collected, classified all our models as carcinomas. We identified invasive features, characteristic of carcinomas, in all tumours collected at the clinical endpoint. Moreover, transcriptomic analysis confirmed this finding as K14 and lvl derived tumours are enriched for the squamous cell carcinoma transcriptional signature (Tumour Specific Keratinocyte TSK population Ji et al 2018). Clarification has been added in the text see from line 153.

We have increased the number of H&E images shown throughout the manuscript, especially those depicting carcinoma lesions (see example in Figure 2). The size and resolution of the H&E and IHC images have been increased. Moreover, higher magnification figures have been

included throughout the manuscript. We believe there was an issue with the file conversion, now resolved; therefore, we are providing a separate document with high-resolution images for the reviewers to assess.

2. In Figure 1 the authors show that HRAS activation synergizes with BRAF inhibitor treatment to induce cSCC formation. In the manuscript the authors claim that this is due to paradoxical MAPK signalling activation (line 87, line 813), the authors should support this claim providing experimental proof of this claim or alternatively, tune down the statement.

Thank you for pointing this out, this suggestion has strengthened the manuscript. As it was also suggested by reviewer 1, we have confirmed MAPK signalling activation by staining the skin with the MAPK downstream markers DUSP6 and pERK in HRAS, het, hom, TPA and in the presence of BRAFi (see Figure 1 and Supplementary Figure 2). This way, we have identified that a threshold level of MAPK signalling activation is required for transformation, which is only achieved with homozygous mutations of HRAS^{G12V}, a combination of heterozygous HRAS^{G12V} and BRAF inhibitors, or BRAF^{V600E}. We have also added graphics to help the reader, see Figures 1g and 2a.

3. In Figure 2 panel E the authors count the number of tumorigenic lesions formed in the different mouse models used. However, for each mouse model a different time point was used, this should be stated in the manuscript text. Otherwise, the authors should study the number of tumours formed in K14:BRAF or K5:BRAF models by resecting the tumours formed until they reach the same timepoint as the Ivl-models. Potentially, the K14 or K5 models might lead to the same number or more tumours formed than the Ivl in a given period of time (the same for all genotypes).

This is a very interesting point. We have clarified the text to acknowledge the present limitation, where K14 and K5 tumours reach the clinical endpoint faster, while Ivl tumours grow more slowly, have more time to accumulate mutant clones, and therefore could explain the multiple tumours. New text: "However, this may be due to the Ivl-driven model having a longer lifespan to develop multiple lesions."

4. It is difficult to interpret Figure 3 in light of the results of Figure 2. In Figure 2 it is shown that in the K14:BRAF V600E model tumours arise with a median tumour onset of 10 days, while in Figure 2 D-F there is no tumour being shown but only wild type skin at Day 8 and Day 15 (missing Day 28 image and quantifications). It would be helpful to understand the rationale of the experiments performed in Figure 2 and why they decided to focus on wild-type skin and not tumorigenic lesions from the K14 model.

We apologise for the lack of clarity. The reviewer is correct; tumours appeared by day 10, so Figure 3 included normal skin adjacent to the tumour for comparison. We can now see how this caused confusion and agree that it does not accurately represent the biology. Thus, the updated version of Figure 3 shows the tumour and the skin immediately adjacent to the tumour to better represent the biology. The day 28 time point was not shown for this model, as it was after the tumour size reached the clinical endpoint, so animals were no longer alive; thus, only control skin was shown. However, this control time point has now been removed to avoid confusion, as it didn't add further insights.

5. In addition, it would be interesting to understand what is the mechanism that is leading to the disappearance of the hyperplastic lesions in the Ivl:BRAF V600E-tdRFP model.

The tumorigenic lesions in the Ivl:BRAF^{V600E}-tdRFP, take longer to develop. However, they do not disappear. The BRAFV600E population expands in the interfollicular epidermis, but the skin remains normal until SOX2 activation reprograms the cells and promotes transformation,

activating the SOX2-stemness transcriptome. This includes increased retention of IVL+ cells in the basal compartment, reduced apoptosis, and tumour-initiating capabilities. To shed some light on this process, we have conducted the following additional analysis:

- Immunohistochemistry analysis at different timepoints, including SOX2, pERK, STAT3, and MYC (Supplementary Figure 6 and Supplementary Figure 7). This data shows that at this early time point, SOX2 is yet not activated, therefore IVL+ population is tumour-resistant.
- IvI:BRAF^{V600E} tumours have apoptotic patches, expressing cleaved CASP3 and PARP (Supplementary Figure 2F), that disappear in tumours overexpressing SOX2 (IvI:BRAFFV600E-SOX2^{LSL}) (Figure 6).
- IvI:BRAFFV600E-SOX2^{LSL} overcome the resistance to transformation by activating CD34 (Figure 6i)

This has been clarified in the discussion as well.

6. In Figure 3 if the authors compare the effect of BRAF V600E activation in the K14 and IvI population using lineage tracing and clonal dynamics analysis. The authors use a non-clonal/saturation dose for the K14 model and a clonal dose for the IvI model. However, in order to be able to directly compare the two models, the authors should be targeting in both models the same number of cells as previously performed (Mascre et al Nature 2012, Sánchez-Danés et al Nature 2016).

We agree with the reviewer and understand the limitation of our approach. We have repeated the lineage tracing experiment in the K14 model using a clonal non-saturation dose as suggested, targeting similar levels as in the IvI:tdRFP model (see Figure 3h).

7. In Figure 4 the authors perform a bulk RNA sequencing of BRAF V600E tumours from the IvI, K14 and K5 models. Are those tumours papillomas or carcinomas and are there differences in the tumour stage between the IvI and K14/K5 analysed?

All samples included in the bulk RNAseq are carcinomas, as shown by their invasive features and their enrichment in TSK signature (see response to point 1). When we compare K14/K5

and Ivl tumours against normal skin ~83% of the upregulated genes are shared (Figure 4d), and the significantly enriched pathways are highly similar (Figure 4e). Moreover, when the tumours are compared against each other, there are only a handful of differentially expressed genes (including Sox2, see Figure 5a), but no other significant transcriptional differences, highlighting similar and comparable levels of tumour stage and progression.

8. The authors perform bulk RNA sequencing in tumour samples, that are not composed of pure tumour cells, but contain both tumour cells and tumour microenvironment. Are the lack of major differences observed between K14&K5 vs Ivl tumours due to an enrichment of tumour microenvironment cell populations that mask the differences in the transcriptional profile of the Ivl vs K14&K5 tumour cells? A possible way to uncover this relevant point would be to FACS sort tumour cells using the BRAF V600E model: RFP (used in Figure 3) and perform bulkRNAseq in those samples.

Thank you for pointing this out. During tumour dissection, every effort was made to isolate the core of the tumour, made of vertical columns of hyperproliferating keratinocytes. Those cores were then taken for RNA isolation and sequencing. Nevertheless, we share the reviewer's concern and proceed to perform digital sorting of the bulk RNAseq tumour samples (see Supplementary Figure 5). As revealed by the CIBERSORTx analysis, the vast majority of the transcriptome comes from keratinocytes derived from the permanent part of the epidermis (~70%). All samples reveal very similar composition profiles, with fibroblasts accounting for ~10% or less of the bulk transcriptome.

Moreover, transcriptomic results have been validated in great depth using immunohistochemistry staining directly in the tumours, and no differences in signalling readouts were identified, except for SOX2 (see examples in Figure 4 and 5, and Supplementary Fig. 6, 7 and 8). We found no evidence to believe that the lack of significant differences is due to sample composition in the bulk RNAseq.

9. In Figure 5 it is shown that SOX2 expression in the Ivl-expressing cells synergizes with BRAF V600E and accelerates tumour formation, but not that" SOX2 overexpression is sufficient to drive tumorigenesis" (line 280). This statement should be corrected, as it is not an accurate description of the data presented.

We have now reworded this statement. Example corrections include: "The IVL+ tumour-resistant basal population depends on SOX2 for transformation", "SOX2 accelerates transformation in the IVL+ tumour-resistant population" and "SOX2 overexpression in combination with MAPK activation renders the IVL+ tumour-resistant population permissive to cSCC".

10. In order to understand if the effect of Sox2 expression is specific to the Ivl-expressing cells, the authors should also show what happens upon Sox2 overexpression in the K14:BRAF V600E model. It would be really interesting to understand if Sox2 overexpression promotes the progression of papillomas to cSCC in the BRAF model.

Thank you for the suggestion. Throughout the paper, we have demonstrated that K14-derived tumours do not express SOX2 (see Figure 5 and Supplementary Figure 7). Additionally, SOX2 KO had no effect on tumour growth or onset in this model (Figure 6). Nevertheless, these tumours grow very quickly, appearing within 7-14 days (Figure 6b). Additionally, all tumours grown from this model were already classified as carcinomas. We do agree that this would be an interesting experiment, unfortunately, due to the time for the revision, we were unable to generate these colonies in a timely manner for his resubmission.

11. The use of “Ivl- expressing cells” throughout the manuscript instead of “tumour-resistant population” is more accurate and might not lead to confusion.

Thank you for the suggestion. We have now adopted “IVL+ tumour-resistant population” through the paper.

12. The title does not describe the current data and should be revised.

We have modified the title from “SOX2 empowers a rapid tumorigenic programme from the tumour-resistant population in the skin” to “SOX2 confers tumour permissiveness in a specific skin progenitor population”

Other specific comments:

1- In Figure 1 the scheme in A is misleading, it seems that the animals are both K14CreER and Ivl-CreER. Same comment for Figure 2A and Figure 3A. Please use the same strategy depicted in Figure 6 A.

This has been corrected. Thanks.

2. Abstract, 3rd sentence. “How the skin tolerates these mutations in different populations... is poorly understood” this statement is not correct as numerous studies from Blanpain lab and Lowry lab among others have contributed to the understanding of the effect of different mutations in cSCC formation.

We agree with the reviewer that multiple labs have done excellent research in the field. For that reason, we have cited previous relevant research to the best of our knowledge. However, normal-looking skin accumulates a great deal of oncogenic-driving mutations without transforming. Its oncogenic resilience compared to other organs is both fascinating and intriguing. The mechanisms that it deploys to keep tumour-free are still incomplete. To our knowledge, this is the first transcriptome comparison between cSCC tumours driven by the K14+ and IVL+ populations. Additionally, it is the first time that SOX2 is linked only to tumours developing from the IVL+ population, rendering it permissive to transformation.

Nevertheless, this paragraph has been changed to: “Over time, multiple oncogenic and tumour suppressor events accumulate. How the skin tolerates these mutations in different populations, and how cutaneous squamous cell carcinoma (cSCC) emerges from mutant clones, remains elusive.”

3. Introduction. Line 66, it should be described that the Ivl and K14 cell populations show different tumour initiating abilities (Sánchez-Danés et al Nature 2016), as this point is really relevant for the present study.

Thank you for the suggestion. This information has now been added in the introduction (Line 38): “These cell populations also differ in their ability to initiate basal cell carcinoma (Sanchez-Danes et al, 2016; Canato et al, 2025).”

4. Figure 4 A, is there a reason for the big dispersion observed within the samples of the control group?

A 9% dispersion is relatively small compared to the 42.6% dispersion between normal skin samples and tumours. Although every attempt has been made to reduce biological variability (i.e same genetic background, same age, littermates), this 9% variation is attributed to biological differences observed in different mice.

5. Figure 4C, K14_K14 in the title should be corrected for K5_K14

This has been corrected. Thanks.

REVIEWER#1 COMMENTS

We thank the reviewer for taking the time to give constructive feedback to improve our manuscript. Below is a point-by-point answer.

1. The authors state in the second sentence of their abstract that “rapid cSCC development occurs in melanoma patients treated with BRAF inhibitors. To model this in mice, we induced MAPK hyperactivation in two epidermal populations: stem-like (K5/K14) and differentiation-committed (IVL+) cells.” It is unclear how these mice would model BRAF inhibitor treatment, whether the mice still require treatment with BRAF inhibitors, and other related details.

The original paragraph has been corrected to increase clarity as follows:

“Rapid cSCC development occurs in melanoma patients treated with BRAF inhibitors (BRAFi), through paradoxical MAPK activation. We modelled this in mice using two complementary approaches; introducing HRAS^{G12V} combined with BRAFi treatment to mimic paradoxical MAPK activation and introducing BRAF^{V600E}, a potent driver of MAPK hyperactivation that does not require further treatment. These mutations were targeted to two basal epidermal populations: stem-like (K5/K14+) and differentiation-committed (IVL+) cells.”

2. The abstract also refers to SOX2 as a pioneer factor. Although this may not be incorrect, this paper does not address a possible pioneer factor function of SOX2. It appears again in line 416. Although a function for SOX2 as a pioneer factor in SCCs has not been demonstrated, Sastre Perona et al, Cell Stem Cell 2019, showed that SOX2 and PITX1 bind to chromatin that is accessible in cSCC but not epidermal progenitor cells. The authors should include this in their discussion and cite this work when arguing a potential role of SOX2 as a pioneer factor that could rewire the epigenome.

We thank the reviewer for this clarification. While SOX2 has been implicated as a super pioneer transcription factor in cellular reprogramming (Vanzan et al 2021; DOI: 10.1038/s41467-021-23630-x), our study indeed does not directly test pioneer activity in cSCC.

We have removed the “pioneer” word from the abstract, and the suggested reference has been added in the discussion as follows:

“In the context of SCC, a bi-stable transcriptional network, involving SOX2, TRP63 and PITX1, has been proposed to promote cell proliferation and to inhibit differentiation by acting on KLF4³⁷. This network may enable tumour cells to switch between proliferation and self-renewal or differentiation and keratin pearl formation.”

3. Line 71-72: .. revealing that tumors retain the transcriptional memory of their cell of origin. How so? It is unclear what the authors refer to with this sentence, and it is unclear from the results which mechanisms are retained and which are not. The authors should be clearer.

We have rewritten this paragraph to increase clarity as follows:

“Our findings demonstrate the different susceptibility of basal populations to cSCC transformation and reveal that SOX2, highly expressed in ~25% of human cSCC and known for its ability to induce stem cell characteristics²⁴⁻²⁶, renders the otherwise tumour-resistant IVL+ population susceptible to oncogenic transformation.”

4. lines 137-143: Why does their IVL-BRAF model develop tumors at multiple sites on the body even without TAM treatment, whereas the K14 model develops tumors only at the spot that was treated? Is this due to leaky or unspecific expression? The authors should explain.

Just to clarify the manuscript does not mention any spontaneous tumours (without TAM treatment) as these are not seen in the IVL model.

Although the TAM induction regimen is applied topically to the back of the skin, it is absorbed and distributed systemically so that any IVL+ cell will express the mutation. Additionally, the additional sites where the IVL-BRAF model develops tumours are epithelial (lips, paws, ears) and common sites of intense grooming. Therefore, they are not considered unspecific and emerging tumours are also cSCC. The same applies to the K14 model; however, due to this “tumour-prime” population transforming so fast, mice reach the end point (due to tumour size) within 10-15 days, and there is no time for tumours to emerge at other body sites. We hypothesise that if primary K14 tumours were resected, additional tumours would also appear in the lips, paws and ears as in the IVL model. However, the tumour body site is not the primary focus of the paper.

This sentence is in the manuscript for clarification:

“However, this may be due to the Ivl-driven model having a longer time to develop multiple secondary lesions.”

5. The authors refer to their tumors throughout as cSCC, but it is unclear whether some of their tumors are benign papillomas.

As Reviewer#2 suggested in the previous revision, we have carefully analysed all the H&E histology collected and classified all our models as carcinomas. We identified invasive features, characteristic of carcinomas, in all tumours collected at the clinical endpoint. Moreover, transcriptomic analysis confirmed this finding as K14 and Ivl derived tumours are enriched for the squamous cell carcinoma transcriptional signature (Tumour Specific Keratinocyte TSK population Ji et al 2018).

We have increased the number of H&E images shown throughout the manuscript, especially those depicting carcinoma lesions (see example in Figure 2).

Clarification in the text, see from line 171:

“Detailed histopathological analysis showed distinctive features of cSCC. These included enlarged interfollicular epidermis and cornified layers with an aberrant accumulation of extracellular keratin (keratin pearls), nuclear atypia, and incomplete maturation of keratinocytes (parakeratosis) as the keratinocytes in the cornified layer retain their nuclei (Fig. 2g-i).”

6. The authors show that SOX2 appears first in differentiation-committed IVL+ cells. Sastre Perona et al, Cell Stem Cell 2019 also reported SOX2 expression to first appear in some suprabasal papilloma cells, before it appeared in most basal cSCC cells. The authors should reference this work.

Thank you for the suggestion. We have added that reference in the introduction and discussion (reference 27 and see response to point 2).

7. lines 240-244: The bulk RNA-seq data must be viewed with caution, as the authors have analyzed total tumour tissue rather than tumour epithelial cells in this study.

This clarification has been added to the text:

“Given the similarities in histology, we next conducted transcriptional profiling of the **total tumour tissue** to identify potential differences and vulnerabilities in these tumours. Therefore it is possible that changes identified here might also reflect changes in epithelial/stroma ratios”.

“It is possible that changes identified here might also reflect changes in epithelial/stroma ratios. Therefore, to validate the transcriptional findings in the epithelial cells, we evaluated protein levels by then performed IHC in tissue slides.”

8. lines 270-275: could identify markers of the cell of origin of cSCC tumours..... SOX2 emerged as a key transcriptional difference, being highly enriched in IVL This paragraph is misleading because SOX2 is not expressed in the cell of origin or normal skin epithelial cells, as reported by several studies. The authors may want to refer to the IVL lineage rather than the cell of origin to avoid confusion.

We appreciate the reviewer’s comment and understand the potential for confusion. The text has been modified as follows:

“We next assessed whether transcriptional profiling could identify potential markers to distinguish tumours arising from different cells of origin.”

9. Why is SOX2 referred to as a super pioneer factor and not a pioneer factor? This terminology seems odd.

This novel terminology refers to the ability of some pioneer transcription factors, including SOX2, to actively induce DNA demethylation at methylated binding sites (see Vanzan et al 2021, doi.org/10.1038/s41467-021-23630-x).

This is clarified in the following sentence:

“SOX2 has been identified as a super pioneer transcription factor capable of binding to closed chromatin through its ability to interfere with the maintenance of DNA methylation²⁴.”

10. line. 286: “We did not detect SOX2 expression or CD34 in either of the models at earlier time points.” This sentence seems out of place in this paragraph. However, it supports my point (8) that the authors shouldn’t refer to the cell of origin if SOX2 is only detected at the experimental endpoint but not at earlier time points.

We understand the potential for confusion. Indeed, we did not detect SOX2 and CD34 at the early time point chosen because the skin was still histologically normal. SOX2 and CD34 are only expressed once transformation is underway and lesions begin to appear, marking a population of tumour initiating cells as previously described in the reference suggested by the reviewer earlier (Boumahdi et al, 2014). Nevertheless, in Figure 6k and Supplementary Fig.9k, we captured tumours emerging expressing SOX2 and CD34 in tumour initiating cells (see snippets below).

This sentence has been clarified as follows:

“We did not detect SOX2 expression or CD34 in either of the models at earlier time points, while the skin remained histologically normal”.

11. Lines 291-299 are convoluted and distractive. It starts by wanting to highlight the cell-specific role of SOX2 and then discusses SOX9, and ends up with a statement on differences in mRNA and protein of KRT4 and KRT13.

We understand the reviewer concern and have significantly shorten the mentioned paragraph as follows:

“We also assessed the expression of SOX9, a distinct regulator of chromatin accessibility that controls hair follicle stem cell fate and is activated during tumorigenesis³⁶. In contrast to SOX2, SOX9 show not cell specificity, as it expression expanded from hair follicles in normal skin to the basal layer in tumours across all models, regardless of cell of origin (**Supplementary Fig. 8a**). The suprabasal markers KRT4 and KRT13, commonly expressed in skin homeostasis, were also

differentially upregulated in the tumours derived from the IVL:BRAF^{V600E} model (Fig. 5a). However, we found no substantial differences at protein level across models (Supplementary Fig. 8b). ”

12. Line 356: “CD34 expression, marking a population of tumor-initiating cells, was observed from day 15..... “ This paper doesn’t test whether CD34-positive cells are tumor-initiating, and previous studies have demonstrated that CD34 expression is dynamic, and CD34+/CD49f+ and CD34-/CD49f+ cells form tumors at the same frequency, and their daughter tumors contain CD34+ and CD34- SCC cells. The interconversion between different cell states in SCCs has also been demonstrated with other cell surface markers and functional features. Therefore, it’s unclear what point the authors want to make.

We appreciate the reviewer’s insight. We intended to highlight that CD34 and SOX2 are activated in transforming cells as shown in figure 6k and supplementary 9k. SOX2 and CD34 mark a population previously associated with tumor-initiating potential in SCC (Boumahdi et al., 2014). To avoid overstatement, we have revised the sentence to:

“CD34 expression was also seen from day 15 and maintained in tumours consistent with a population of tumour initiating cells²⁴ (Supplementary Fig. 9c).”

13. Line 366: .. the apoptosis hallmark described here was reported by Boumahdi et al Nature 2014. The authors should add a reference to their finding as it confirms Boumahdi’s findings.

We added the reference one more time at the suggested location.

14. line 371-372: “... IVL+ derived tumors are indistinguishable from those originating from a K14/5+ stem like population.” - This statement must be wrong as the authors argue throughout that IVL+ express and depend on SOX2, whereas K14/5 derived tumors do not.

We thank the reviewer for this point. We agree that “indistinguishable” could be misleading. We have modified the text as follows:

“The resulting IVL+ derived tumours share most histological features and a core transcriptional programme of transformation with those originating from a K14/K5+ stem-like population (Fig. 7).”

15. line 400: The sentence ends with shown here, but it is unclear what core set of skin transformation pathways the authors are referring to and where they are shown in this paper.

The sentence has been amended as follows:

“Despite the diversity of oncogenic events across these models, they converge on the core set of skin transformation pathways including wound-healing, embryonic and tumour-specific keratinocyte signatures, and MYC and hypoxia/STAT3 activation.”

16. line 432 is an overstatement: This manuscript shows SOX2 promotes/is required for cSCC development in an IVL-BRAF model consistent with previous reports in lung cancer (Xu et al, Genes and Development 2014), but it does not show that the cells are being reprogrammed. To support this claim, the authors would need to profile the epigenomes of the respective cells of origin and their tumors to demonstrate that the cells were reprogrammed.

This sentence has been modified as follows:

“However, committed progenitors can trigger skin tumorigenesis once they re-acquire stemness features, induced by SURVIVIN expression in basal cell carcinoma⁶ or by SOX2 in cSCC, as shown here.”

17. line 442: “... preventing delamination and reducing apoptosis” The authors should include references to Siegle et al Nat. Comm 2014 shows that SOX2 affects symmetric versus asymmetric renewal, which is presumably similar to delamination here, and Boumahdi should be referenced regarding reduced apoptosis.

The suggested references have been added once more at the suggested location.